# DRESS: Disentangled Representation-based Self-Supervised Meta-Learning for Diverse Tasks

## Abstract

Meta-learning represents a strong class of approaches for solving few-shot learning tasks. Nonetheless, recent research suggests that simply pre-training a generic encoder can potentially surpass meta-learning algorithms. In this paper, we hypothesize that the reason meta-learning fails to stand out in popular few-shot learning benchmarks is the lack of diversity among the few-shot learning tasks. We propose *DRESS*, a task-agnostic Disentangled REpresentation-based Self-Supervised meta-learning approach that enables fast model adaptation on highly diversified few-shot learning tasks. Specifically, DRESS utilizes disentangled representation learning to create self-supervised tasks that can fuel the meta-training process. We validate the effectiveness of DRESS through experiments on datasets with multiple factors of variation and varying complexity. The results suggest that DRESS is able to outperform competing methods on the majority of the datasets and task setups. Through this paper, we advocate for a re-examination of how task adaptation studies are conducted, and aim to reignite interest in the potential of meta-learning for solving few-shot learning tasks via disentangled representations.

## 1 Introduction

Few-shot learning (Wang et al., 2020) emphasizes the ability to quickly learn and adapt to new tasks, and is regarded as one of the trademarks of human intelligence. In the pursuit of few-shot learning, meta-learning approaches have been widely explored (Finn et al., 2017; Snell et al., 2017; Ravi & Larochelle, 2017), as they allow models to *learn-to-learn*. However, multiple recent studies (Tian et al., 2020; Dumoulin et al., 2021; Shen et al., 2021; Shu et al., 2023) suggest that a simple *pre-training and fine-tuning* approach is sufficient to support highly competitive performance in few-shot learning tasks. Specifically, a generic encoder is trained with a self-supervised loss on a unified dataset that aggregates samples (with their targets dropped) from all available training tasks. A linear layer is added on top of the encoder and is fine-tuned using few-shot support samples to adapt to new tasks. Pre-training and fine-tuning neglects two crucial sources of information: identities of individual meta-training tasks and distinctions between them; and labels in meta-training tasks. Yet, pre-training and fine-tuning has been shown to achieve better results than meta-learning. This finding is unexpected, and perhaps even puzzling, as it implies that information about training tasks and their labels may be irrelevant to achieving high learning performance.

We hypothesize that this finding can be attributed to the lack of *task diversity* in many popular few-shot learning benchmarks. For instance, in canonical few-shot learning datasets such as Omniglot (Lake et al., 2011), *mini*ImageNet (Vinyals et al., 2016), and CIFAR-FS (Bertinetto et al., 2019), the distinct tasks differ solely in that their targets belong to non-overlapping sets of object classes. In essence, these tasks all share the same nature: main object classification. Hence, there is one degenerate strategy for solving all these tasks simultaneously: compare the main object in the query image to the main objects in the few-shot support images, and assign the class label based on similarity to support images. This strategy can be achieved through pre-training with contrastive learning using common image augmentations like rotation and cropping which preserve the semantics of the main object, while discarding factors such as orientation and background (Balestriero et al., 2023). Given the shared nature of tasks on these benchmarks, it is not surprising that a single pre-trained encoder can perform competitively against meta-learning methods.

To rigorously challenge a model's adaptation ability, we advocate for the establishment of few-shot learning benchmarks that include tasks with fundamentally distinctive natures. Specifically, we consider tasks beyond main object classification, such as identifying object orientation, background color, ambient lighting, or attributes of secondary objects. In addition, models should be *agnostic* to the nature of the evaluation tasks. Such setups can reveal the model's true capacity to learn strictly from the few-shot samples, with *task identification* as an essential learning component. Furthermore, we highlight a key consequence of high task diversity: when meta-testing tasks differ significantly in nature from meta-training tasks, the labels in meta-training tasks may provide misleading guidance to the model, towards premature fixation on a narrow perspective of the input data. Recognizing this issue, we reaffirm the preference of *self-supervised* meta-learning over supervised meta-learning.

For effective meta-learning under high task diversity, we bridge the idea of disentangled representation learning with self-supervised meta-learning in a single framework referred to as *DRESS — task-agnostic Disentangled REpresentation-based Self-Supervised meta-learning*. Specifically, we utilize an encoder trained to compute disentangled representations, and extract latent encodings of the inputs. We then semantically align these latent representations across all inputs. Within this aligned latent space, we perform clustering independently on each disentangled latent dimension, and use the resultant cluster identities to define pseudo-classes of the inputs. Finally, we construct a set of self-supervised few-shot classification tasks based on these pseudo-classes from each latent dimension. With the disentangled latent dimensions representing distinct attributes and factors of variation within the inputs, the constructed few-shot learning tasks are highly diversified. Using these tasks for meta-training, the model can learn to adapt quickly to unseen tasks, regardless of the task nature. In addition, we propose a quantitative task diversity metric based on class partitions. Our metric is directly defined on the input space instead of any learned embedding space, therefore allowing fair and independent comparisons between tasks of distinct semantic natures.

We conduct extensive experiments on image datasets containing multiple factors of variation, beyond the main object's class, and spanning different levels of complexity and realism. To ground our results, we establish three supervised meta-learning baselines that have differing levels of ground-truth information. These supervised baselines not only serve as upper bounds on performance, but also expose the negative effects of learning from labels when the natures of tasks are mismatched. Our results suggest that DRESS enables few-shot learning performance that can surpass existing methods, and approaches the upper bound of supervised baselines under many experimental setups.

Our main contributions can be summarized as follows:

- We identify the lack of task diversity in few-shot learning benchmarks, explaining why pre-training and fine-tuning can seem to outperform meta-learning.

- We develop few-shot learning benchmarks with more diversified tasks for rigorous evaluation.

- We propose DRESS, a method for creating diverse tasks that enable self-supervised meta-learning with disentangled representations.

- We introduce a task diversity metric based on task class partitions directly over the input space.

## 2 Related Works

**Meta-Learning *vs.* Pre-training and Fine-tuning**   There has been a large volume of supervised meta-learning research on the general few-shot learning problem (Finn et al., 2017; Snell et al., 2017; Lee et al., 2022; Song et al., 2022; Kim & Hospedales, 2024; Wu et al., 2025; Wang et al., 2025). Among them, several works focus on incorporating the information of task distribution into the learning process. Specifically, Wu et al. (2025) explores using the task identity information for contrastive learning on the space of model weights. Meanwhile, Wang et al. (2025) focuses on the problem setting of continuous task adaptation, where out-of-distribution tasks imposed by an adversarial task generator are considered in the meta-testing stage. On the other hand, researchers have also explored unsupervised or self-supervised meta-learning (Hsu et al., 2019; Khodadadeh et al., 2019; 2021; Lee et al., 2021; Jang et al., 2023; Pachetti et al., 2024). Notably, CACTUS (Hsu et al., 2019) proposes a task construction approach using an encode-then-cluster procedure.

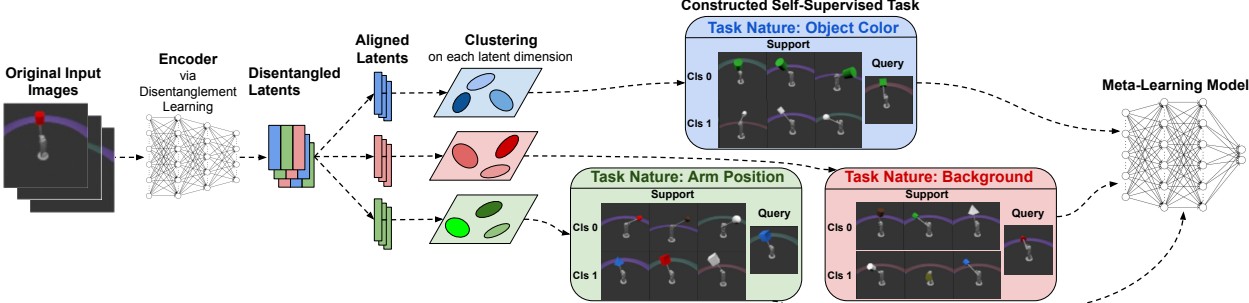

Figure 1: DRESS creates diversified self-supervised meta-training tasks through disentanglement learning. Images are first encoded into disentangled latent representations. The latent representations are then semantically aligned across dataset. Clusters are formed on each latent dimension individually. Pseudo-classes are sampled from these clusters to construct self-supervised classification tasks. Each disentangled latent dimension corresponds to a set of tasks with its unique nature.

Meta-GMVAE (Lee et al., 2021) models the dataset using a variational auto-encoder with a mixture of Gaussians as prior, and matches latent modalities with class concepts. Studies including (Khodadadeh et al., 2019; Jang et al., 2023) use image augmentations to create samples for pseudo classes to meta-train the model. Although promising results are obtained on standard few-shot learning benchmarks, these works do not explicitly address the issue of task diversity, nor its effect on fast adaptation performance.

Recent studies (Tian et al., 2020; Dumoulin et al., 2021; Shen et al., 2021; Shu et al., 2023) state that the simple approach of pre-training a generic encoder followed by fine-tuning can show superior performance compared to meta-learning. Specifically, the input samples from all available meta-training tasks are aggregated into a large dataset, with task identities completely ignored. An encoder is then trained on this large dataset using supervised or self-supervised training techniques (*e.g.*, contrastive learning (Chen et al., 2020)). When adapting to any meta-testing task, a linear classification layer is added and fine-tuned on top of the encoder over the support samples.

**Task Diversity** The main obstacle to investigating the task diversity is the difficulty of quantifying it. Existing research measures task diversity either via establishing a shared embedding space (Achille et al., 2019; Kumar et al., 2022), or through projection mappings from the input to output spaces (Sui et al., 2024). Recently, Miranda et al. (2023) conducted thorough experiments suggesting that existing meta-learning methods can show very slight improvements over the pre-training and fine-tuning approach on tasks with higher *Task2Vec* diversity coefficients (Miranda et al., 2022). Nonetheless, the intuition behind the link between task diversity and the performance of few-shot learning has yet to be discussed. Similarly, no meta-learning approach has explicitly exploited the idea of diversifying meta-training tasks for boosting the fast adaptation ability of a model.

**Disentangled Representation Learning** Disentangled representation learning has been mainly investigated in the context of generative modeling (Higgins et al., 2017; Kim & Mnih, 2018; Singh et al., 2022; Yang et al., 2023; Hsu et al., 2024; Jiang et al., 2023; Yue et al., 2024; Wu & Zheng, 2024), with the objective of learning representations that capture independent factors of variation within the input distribution. For complex images, factors of variations include the main object identity, as well as object orientation, background, ambient lighting, view angle, and so on.

## 3 Methodology

We introduce DRESS, our task-agnostic Disentangled REpresentation-based Self-Supervised meta-learning approach. DRESS leverages disentangled latent representations of input images to construct self-supervised few-shot learning tasks that power the meta-training process. The multi-stage diagram and pseudo code of DRESS are provided in Figure 1 and Algorithm 1 respectively.

### 3.1 Encoding Disentangled Representations

First, all images available for meta-training are collected, and used to train a general purpose encoder with the objective of producing disentangled representations (*e.g.*, a factorized diffusion autoencoder (FDAE) (Wu & Zheng, 2024), or latent slot diffusion model (LSD) (Jiang et al., 2023)). We then use the trained encoder to encode each image and obtain its disentangled latent representation, which consists of a set of vectors, one for each identified semantic concept. For the remainder of the paper, we resort to the term *dimension* to refer to individual semantic concepts. We rely on prior information about the dataset to select the appropriate number of latent dimensions to encode (i.e. how many factors of variation are expected based on image structure). However, when such information is not available, the intrinsic dimension of the dataset can be used as a proxy (Loaiza-Ganem et al., 2024; Kamkari et al., 2024).

The notion of *entanglement* is broad and may correspond to various definitions. For example, in $\beta$-VAE (Higgins et al., 2017) and FDAE, the entanglement of latent representations is connected to covariance; in factorVAE (Kim & Mnih, 2018), feature entanglement is quantified statistically as *total correlation*; while for LSD, feature entanglement is translated to relative spatial locations in the image space. As DRESS is compatible with various encoder designs, in Algorithm 1, we use *entanglement(·)* to denote a general notion of entanglement, with its specific definition depending on the selected encoder.

### 3.2 Aligning Latent Dimensions

After collecting the disentangled representations for all the training images, we align the latent dimensions of representations across images so that a given dimension conveys the same semantic information across all images (*e.g.*, main object color, object orientation, background color, lighting condition). For instance, some encoders (Locatello et al., 2020; Singh et al., 2022; Jiang et al., 2023) disentangle attributes by applying multiple attention masks over each image. For such latent spaces, we can align latent features by aligning the attention masks in spatial dimensions. Attention masks that are similar in shapes and spatial locations generally focus on the same semantic elements across images. To align such attention masks, we first preprocess each attention mask by flattening it and normalizing it into a vector on the simplex. We then gather a batch of attention masks and cluster them with K-Means (with the number of clusters equal to the number of attention masks learned on each image). With the obtained attention mask clusters, we reorder the latent representations from all images by the cluster identities of their corresponding attention masks.

### 3.3 Clustering Along Disentangled Latent Dimensions

We perform clustering within each dimension over latent vectors. Since dimensions are disentangled and aligned, clustering each dimension produces a distinct partition of the entire set of inputs that corresponds to one semantic property. Similar to Section 3.1, the number of clusters in this stage is a design choice. To shape the constructed tasks towards higher levels of difficulty, thus encouraging the model to learn data variations on finer levels of granularity, one can increase the number of clusters per dimension.

### 3.4 Forming Diverse Self-Supervised Tasks

Finally, we construct self-supervised learning tasks using cluster identities as *pseudo-class* labels. We create a large number of few-shot classification tasks under each disentangled latent dimension by first sampling a subset of cluster identities as classes, and then sampling images under each class as the few-shot support samples and query samples.

As different dimensions within the disentangled representation depict distinct aspects of the input data, the sets of self-supervised tasks constructed from disentangled dimensions are naturally diversified, requiring distinct decision rules to solve. When using these tasks for meta-training, the model can digest each factor of variation within the data, and therefore learns to adapt to unseen few-shot tasks regardless of their contexts, natures, and meanings.

### 3.5 Selection of the Meta-Learning Algorithm

DRESS is compatible with any conventional meta-learning algorithm for model training. However, not all meta-learning algorithms are well-suited to the highly diversified tasks DRESS generates. In this paper, we pair DRESS with the optimization-based adaptation approach MAML (Finn et al., 2017) because of its

---

**Algorithm 1** DRESS Pipeline on $N$-way $K$-shot tasks

---

1: **Input:** $\{x_i\}_{i=1}^{K_{\text{total}}}$ .
2: Train encoder with disentanglement learning on inputs
$$f_{\text{enc}} : \mathcal{X} \to \mathcal{R}^{J \times L} \quad (J\text{: dimensions}, L\text{: latent size}).$$
3: Obtain disentangled representations
$$\mathbf{f}_i = f_{\text{enc}}(x_i)$$
$$\text{s.t. entanglement}(\mathbf{f}_i^{j_1}, \mathbf{f}_i^{j_2}) \approx 0 \quad \forall i, \forall_{j_1, j_2 \in [J], j_1 \neq j_2}.$$
4: Align latent dimensions by permuting $j$ for each image, so that the semantic information in $\mathbf{f}_i^j$ is consistent across all images.
5: **for** $j \in [J]$ **do**
6:     Cluster $\{x_i\}$ on $\mathbf{f}^j$ to define a partition $P_j$.
7: **end for**
8: **while** not converged **do**
9:     Sample a partition $P \sim \{P_j\}$
10:     Sample $N$ clusters from the partition $\{C_{c_i}\} \sim P$
11:     **for** $c_i \in [N]$ **do**
12:         Sample $K$ datapoints from cluster $C_{c_i}$ as *support* samples, set class labels as $y_i^s = c_i$.
13:         Sample $K$ datapoints from cluster $C_{c_i}$ as *query* samples, set class labels as $y_i^q = c_i$.
14:     **end for**
15:     Perform one meta-learning optimization step.
16: **end while**

---

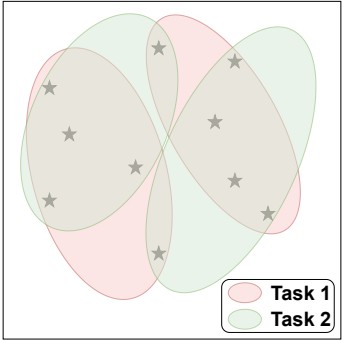 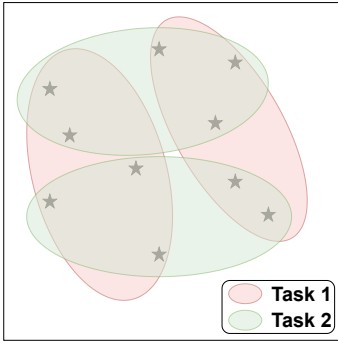

Figure 2: Illustration of class partition-based task diversity. A binary classification task is defined by two ellipses of the same color on an input space. **Left:** Two similar tasks where classes have high overlap among data points. **Right:** Two dissimilar tasks, with less overlap between class partitions.

simplicity and ubiquity in meta-learning benchmarks. See Figure 6 in Appendix A.1 for a general illustration of the meta-learning pipeline. Discussions of pairing DRESS with other popular meta-learning algorithms are in Appendix A.2.

### 3.6 Task Diversity based on Class Partitions

In DRESS, different encoders with different embedding spaces could be used to construct tasks. Correspondingly, we advocate for a task diversity metric that is not tied to any specific embedding space, but is directly linked to the original input space, unlike metrics such as Task2Vec. We introduce a task diversity metric based on task class partitions. Consider two classification tasks defined on the same inputs as in Figure 2. Each task partitions the dataset based on class identities. The similarity between the tasks can be measured by the similarity between their respective partitions.

The mathematical definition of our metric is as follows: consider an input dataset of $K$ data points, $\mathcal{D} = \{\mathbf{x}_i\}_{i=1}^{K}$, and two (potentially multi-class) classification tasks, $T_1$ and $T_2$, defined on $\mathcal{D}$. Assume $T_1$ and $T_2$ both have $N$ classes which can be mapped to $\{c_j\}_{j=1}^{N}$ (if one task has fewer classes, we treat the missing

classes as having zero samples). $T_1$ and $T_2$ can be described by two sets of class labels $\{y_i^1\}_{i=1}^K$ and $\{y_i^2\}_{i=1}^K$, respectively, with one label for each input in $\mathcal{D}$. Equivalently, each task can be represented by a class-based partition of $\mathcal{D}$. For $T_1$, the class partition is denoted as $\mathcal{P}^1 = \{P_{c_j}^1\}_{j=1}^N$, where $P_{c_j}^1 = \{x_i \mid y_i^1 = c_j\}$. Similarly, $\mathcal{P}^2$ represents the class partition for $T_2$.

Our task diversity metric is computed using these class-based partitions. First, we match subsets between $\mathcal{P}^1$ and $\mathcal{P}^2$ to maximize the pairwise overlaps, via methods such as bipartite matching. For each matched pair of subsets, we compute the intersection-over-union (IoU) ratio. Finally, we calculate the average IoU value across all subset pairs across the two partitions. A low average IoU indicates that $\mathcal{P}^1$ and $\mathcal{P}^2$ differ significantly, suggesting that $T_1$ and $T_2$ are relatively diverse tasks. We show pseudocode for the metric in Algorithm 2 in Appendix J. We note that during the step of relabeling the classes, the semantic information of the classes in each task is lost. Therefore, the proposed metric only quantifies task diversity from the function mapping perspective. Nonetheless, learning to jointly solve tasks that are diversified in their input-output mappings (which our metric quantifies) has been shown to enable better adaptation capacity (Sui et al., 2024).

## 4 Experimental Setup

### 4.1 Datasets

We consider curated datasets with controlled factors of variations, as well as complex real-world datasets. For curated datasets, we consider *SmallNORB* (Lecun et al., 2004), *Shapes3D* (Burgess & Kim, 2018), *Causal3D* (von Kügelgen et al., 2021), and *MPI3D* (Gondal et al., 2019), covering a data-complexity spectrum from easy to hard. These datasets include labels for multiple independently varying factors. For real-world datasets, we explore *CelebA* (Liu et al., 2015a) and *LFWA* (Liu et al., 2015b). Details of the factors of variation in each dataset are in Appendix B. To complement our analysis regarding the effect from task diversity within the dataset, we also provide performances on the conventional benchmarks including *Omniglot* (Lake et al., 2011) and *mini*ImageNet (Vinyals et al., 2016).

### 4.2 Implementation Details of DRESS

**Curated Datasets**: For our experiments on SmallNORB, Shapes3D, Causal3D and MPI3D, as well as low-task-diversity datasets Omniglot and *mini*ImageNet, we adopt the FDAE architecture (Wu & Zheng, 2024) for the encoder. We train a FDAE model from scratch on each dataset and use it to encode the images into disentangled representations. The FDAE encoder computes a content code and a mask code for each visual concept. We regard this pair of codes as two independent latent dimensions, which enables DRESS to construct self-supervised tasks that capture information ranging from semantic and contextual cues to spatial relations and structure within images. The number of visual concepts that we adopt for each dataset is provided in Appendix D. When using FDAE as the encoder, no explicit computation is required for the latent alignment stage in DRESS. Since FDAE employs deterministic convolutional neural networks, each output head of the encoder computes a fixed semantic mapping. Therefore, the latent dimensions are inherently organized in a consistent semantic order. This allows us to proceed directly to clustering after encoding all images. We then perform individual latent dimension clustering and pseudo-class construction, with details in Appendix D.

**Real-World Dataset**: For CelebA and LFWA experiments, we adopt the LSD encoder (Jiang et al., 2023) trained from scratch to demonstrate DRESS's flexibility in adapting to representations from various encoder architectures. The LSD encoder utilizes slot attention to learn disentangled latent representations by computing visual *slots*, with each slot attending to different regions of the image through a learned attention mask. Due to the stochastic nature of slot attention, the order of the slots varies across images, requiring explicit latent alignment before clustering. We align the latent dimensions by clustering on the attention masks from each slot, as discussed in Section 3.2. We provide detailed description on this alignment procedure and visualizations in Appendix F. After alignment, we perform clustering and form pseudo-classes for self-supervised meta-training tasks.

### 4.3 Baseline Methods

**Supervised Meta-Learning**: We implement three variations of supervised meta-learning baselines with increasingly relevant information about ground-truth factors:

- *Supervised-Original*: Only use the ground-truth factors that do not define meta-testing tasks to create supervised meta-training tasks.

- *Supervised-All*: Use all the ground-truth factors to create supervised meta-training tasks.

- *Supervised-Oracle*: Only use the ground-truth factors that define meta-testing tasks to create supervised meta-training tasks.

These methods progressively increase the relevancy of information available to the model, but are increasingly unrealistic. Supervised-Original must learn to generalize from a limited set of ground-truth factors to unknown factors at meta-testing time. Specifically, for Supervised-Original, the ground-truth factors available in meta-training and in meta-testing are mismatched. As a result these ground truth factors can potentially misguide the model causing worse generalization ability. Supervised-All has the most information, but needs to identify task natures and relevant factors, and hence represents the upper bound on performance when the evaluation tasks are agnostic. Supervised-Oracle has perfect knowledge of factors utilized in meta-testing tasks, and represents the ultimate performance upper bound.

**Few-Shot Direct Adaptation (FSDA)**: This represents the lower bound of performance when a model is directly optimized on the support samples from each meta-testing task.

**Pre-training and Fine-tuning (PTFT)**: We implement the pre-training and fine-tuning method as described in (Tian et al., 2020), using SimCLR (Chen et al., 2020) with its standard image augmentations, with details in Appendix C.

**Unsupervised & Self-Supervised Meta-Learning**: We adopt *CACTUS* (Hsu et al., 2019) with two encoders: DeepCluster (Caron et al., 2018) trained from scratch, and off-the-shelf DINOv2 (Oquab et al., 2024). We refer to these baselines as *CACTUS-DC* and *CACTUS-DINO*, with details in Appendix D. We note that as one of the state-of-art vision encoders, DINOv2 is trained at substantially larger scale and offers higher capacity and richer representations than any encoder we used in DRESS. Additionally, we experiment with two recent unsupervised and self-supervised meta-learning approaches: *Meta-GMVAE* (Lee et al., 2021) and *PsCo* (Jang et al., 2023).

We unify the model architecture, meta-training, and meta-testing setups for these methods across all experiments, as detailed in Appendix D. We also emphasize that for each dataset, the same set of images is used for meta-training (either the encoder or the meta-learner model) and for pre-training and fine-tuning, with the only exception being the DINOv2 encoder used in the CACTUS baseline, which has been extensively trained on much larger training sets. Essentially, the information available for training is identical in each of the competing methods.

### 4.4 Meta-Training & Meta-Testing Task Setups

For meta-testing, we construct few-shot learning tasks based on the selection of a *subset* of the attributes with ground-truth labels from each dataset. Consequently, the natures and levels of difficulty of the tasks are determined by this subset of attributes. Given the subset of attributes selected, the meta-testing tasks are created using the ground-truth labels, similar to (Hsu et al., 2019). First, we randomly pick a few attributes from the attribute subset, and define two distinct value combinations on those attributes. Images whose attributes match the first value combination are assigned to the positive class, while those matching the second combination are assigned to the negative class. For the three supervised meta-training baselines, we also create supervised meta-training tasks following the same procedure. Details on the subsets of attributes for supervised meta-training tasks and meta-testing tasks are provided in Appendix B for each dataset. For meta-training, we construct *2-way 5-shot* few-shot learning tasks. While for meta-testing, we also experiment with *2-way 10-shot* tasks to better examine the adaptation ability of each method.

We create multiple meta-testing configurations for the two most complex datasets, MPI3D and CelebA, by varying how attributes are grouped. For MPI3D, we define two few-shot learning setups: *MPI3D-Easy*,

Table 1: Few-shot classification accuracies on curated datasets, with each trial conducted over 1000 meta-testing few-shot learning tasks.

| Method | SmallNORB | | Shapes3D | | Causal3D | | MPI3D-Easy | | MPI3D-Hard | |
|---|---|---|---|---|---|---|---|---|---|---|
| | 5-Shot | 10-Shot | 5-Shot | 10-Shot | 5-Shot | 10-Shot | 5-Shot | 10-Shot | 5-Shot | 10-Shot |
| Supervised-Original | 61.9% ±0.8% | 65.3% ±1.7% | 62.0% ±1.5% | 70.3% ±1.7% | 52.1% ±0.3% | 52.9% ±0.3% | 57.8% ±0.5% | 64.6% ±1.4% | 63.3% ±1.3% | 65.0% ±1.3% |
| Supervised-All | 79.6% ±0.3% | 80.8% ±0.4% | 99.9% ±0.0% | 100.0% ±0.0% | 88.8% ±1.0% | 90.4% ±1.2% | 99.3% ±0.3% | 99.9% ±0.0% | 91.0% ±1.7% | 94.4% ±0.5% |
| Supervised-Oracle | 80.2% ±0.4% | 82.0% ±0.2% | 100.0% ±0.0% | 100.0% ±0.0% | 93.5% ±0.2% | 94.4% ±0.3% | 100.0% ±0.0% | 100.0% ±0.0% | 99.4% ±0.1% | 99.7% ±0.1% |
| FSDA | 73.9% ±0.9% | 74.4% ±0.8% | 65.7% ±2.0% | 87.8% ±0.6% | 66.9% ±0.9% | 67.8% ±3.1% | 60.6% ±0.3% | 97.4% ±0.1% | 62.3% ±0.3% | 66.7% ±0.9% |
| PTFT | 58.0% ±1.9% | 61.9% ±1.1% | 57.9% ±2.2% | 71.6% ±0.2% | 55.6% ±0.2% | 57.2% ±0.6% | 92.9% ±0.5% | 84.8% ±0.4% | 79.5% ±0.8% | 94.1% ±0.3% |
| Meta-GMVAE | 68.6% ±0.7% | 73.9% ±0.3% | 59.1% ±1.7% | 59.6% ±0.9% | 59.2% ±0.8% | 63.9% ±0.6% | 99.4% ±0.1% | 99.2% ±0.3% | 50.0% ±0.3% | 50.6% ±0.2% |
| PsCo | 74.2% ±0.4% | 74.3% ±0.6% | **97.6%** ± 0.6% | 91.4% ±0.5% | 70.8% ±0.5% | 76.0% ±0.5% | 83.5% ±2.0% | 96.7% ±0.9% | 79.5% ±0.7% | 89.6% ±0.3% |
| CACTUS-DC | 75.8% ±0.4% | 76.3% ±0.4% | 86.8% ±0.7% | 93.5% ±0.4% | 65.7% ±0.4% | 69.7% ±0.7% | 85.0% ±0.6% | 92.6% ±0.7% | 72.8% ±1.0% | 79.2% ±0.4% |
| CACTUS-DINO | 62.8% ±0.8% | 66.9% ±1.0% | 80.6% ±0.2% | 89.3% ±0.0% | 53.9% ±0.5% | 56.0% ±0.3% | 94.4% ±0.4% | 97.7% ±0.3% | 81.9% ±0.4% | **89.0%** ± 0.5% |
| **DRESS** | **78.1%** ± 0.4% | **79.1%** ± 0.2% | 93.1% ±0.2% | **97.1%** ± 0.4% | **76.4%** ± 0.4% | **80.4%** ± 0.2% | **99.9%** ± 0.0% | **100.0%** ± 0.0% | **85.0%** ± 0.5% | 88.4% ±0.4% |

where the tasks focus on identifying the background and camera height; and *MPI3D-Hard*, where the tasks focus on horizontal and vertical robot arm angular positions. For CelebA, we define three few-shot learning setups: *CelebA-Hair*, where the tasks focus on all attributes relevant to the person's hair; *CelebA-Primary*, where the tasks focus on primary facial attributes or features; and *CelebA-Random*, where the tasks are constructed from a random subset of attributes.

Lastly, to examine the ability of *cross-domain adaptation*, we adapt each model trained under the CelebA dataset onto the few-shot learning tasks created based on LFWA under a subset of primary attributes. We refer to this cross-domain adaptation setup as *LFWA-Cross-Domain*. *Supervised-Oracle* is no longer a valid baseline under this setup. As we adapt from CelebA to LFWA, the set of attributes are changed. Therefore, there is no oracle information on the meta-testing attributes.

## 5 Results & Analysis

### 5.1 Experimental Results on Curated Datasets

We present the few-shot classification accuracies in Table 1 for all the curated datasets.[1] DRESS consistently achieves the best few-shot adaptation performance among unsupervised or self-supervised methods under the most setups with just two exceptions. Supervised-Original is unimpressive, indicating that meta-training targets could mislead a supervised model when adapting to highly diversified tasks, as we hypothesized in Section 3. In contrast to Tian et al. (2020), pre-training and fine-tuning is not on par with meta-learning approaches, due to the more challenging and diverse tasks we benchmark on. CACTUS shows varying results across datasets with different encoders, reflecting the importance of the latent representations. As DRESS uses disentangled representation learning to construct diversified pre-training tasks, it obtains superior results across these datasets and task setups. We provide visualizations of two tasks constructed by DRESS in Figure 3, and additional visualizations in Appendix H. Furthermore, we also provide the few-shot classification accuracies on the low-task-diversity benchmarks in Appendix G. As shown by the results, when the dataset and its corresponding tasks enter the low-task-diversity regime, the Pre-training and Fine-tuning approach becomes highly competitive to the meta-learning approaches, confirming our earlier hypothesis on the importance of the task diversity when evaluating few-shot learning performances.

---

[1]All reported results show mean and standard deviation over 4 trials under random seeds.

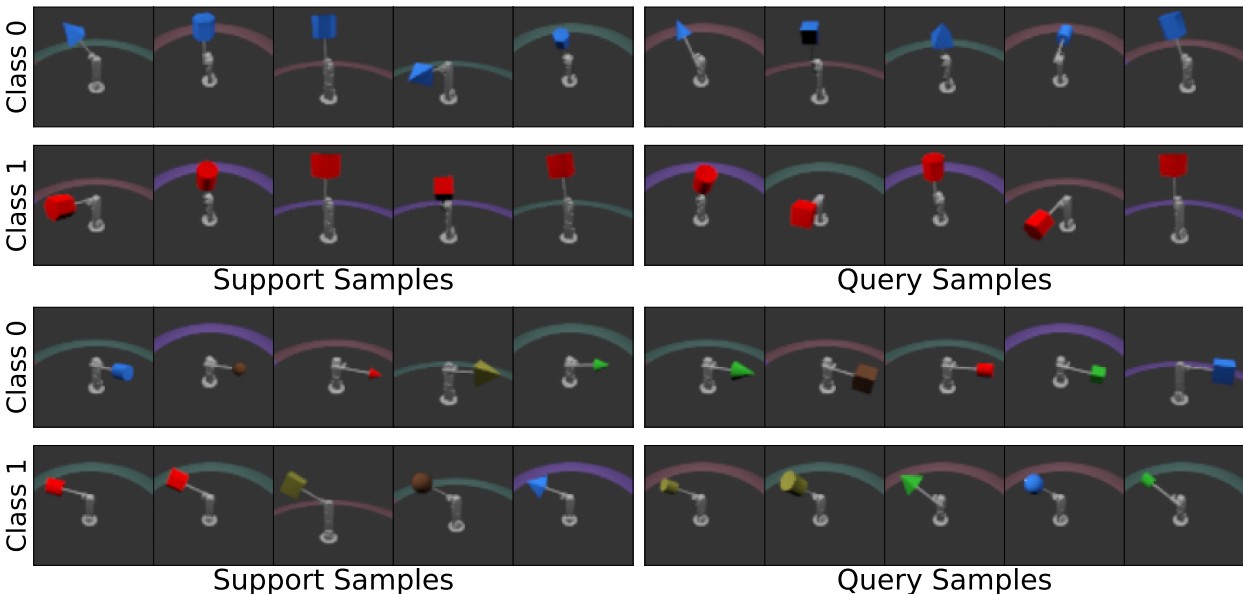

Figure 3: Two self-supervised tasks constructed by DRESS on MPI3D. **Top:** The task focuses on classifying the object color. **Bottom:** The task focuses on identifying the robot arm angle.

Table 2: Few-shot classification accuracies on the realistic CelebA dataset and LFWA dataset for cross-domain adaptation, with each trial conducted over 1000 meta-testing few-shot learning tasks.

| Method | CelebA-Hair | | CelebA-Primary | | CelebA-Random | | LFWA-Cross-Domain | |
| | 5-Shot | 10-Shot | 5-Shot | 10-Shot | 5-Shot | 10-Shot | 5-Shot | 10-Shot |
|---|---|---|---|---|---|---|---|---|
| Supervised-Original | 68.9% | 71.8% | 77.0% | 79.9% | 81.9% | 83.9% | 69.1% | 71.6% |
| | ±0.6% | ±0.2% | ±1.1% | ±1.0% | ±0.2% | ±0.2% | ±1.6% | ±1.4% |
| Supervised-All | 79.1% | 81.9% | 88.1% | 89.2% | 85.6% | 87.6% | 73.0% | 75.3% |
| | ±0.2% | ±0.1% | ±0.2% | ±0.5% | ±0.2% | ±0.1% | ±0.3% | ±0.0% |
| Supervised-Oracle | 87.8% | 89.3% | 91.2% | 92.1% | 90.7% | 92.0% | - | - |
| | ±0.3% | ±0.1% | ±0.1% | ±0.1% | ±0.1% | ±0.2% | | |
| FSDA | 63.3% | 63.3% | 69.3% | 70.0% | 57.7% | 56.7% | 61.8% | 62.0% |
| | ±0.2% | ±1.4% | ±0.5% | ±1.0% | ±0.4% | ±1.2% | ±0.9% | ±0.7% |
| PTFT | 59.6% | 62.0% | 67.1% | 70.3% | 65.1% | 68.0% | 62.7% | 65.7% |
| | ±0.3% | ±0.2% | ±0.3% | ±0.2% | ±0.3% | ±0.1% | ±0.1% | ±0.1% |
| Meta-GMVAE | 64.2% | 68.9% | 67.9% | 72.4% | 64.9% | 68.2% | 59.4% | 61.7% |
| | ±0.2% | ±0.1% | ±0.3% | ±0.4% | ±0.2% | ±0.1% | ±0.2% | ±0.1% |
| PsCo | 66.2% | 67.2% | 66.0% | 70.9% | 60.5% | 67.5% | 59.0% | 61.5% |
| | ±0.3% | ±0.8% | ±0.6% | ±0.5% | ±0.4% | ±0.9% | ±0.3% | ±0.5% |
| CACTUS-DC | 67.4% | 70.8% | 71.4% | 75.8% | 62.2% | 66.4% | 63.9% | 67.1% |
| | ±1.0% | ±1.4% | ±0.1% | ±0.8% | ±1.1% | ±1.6% | ±1.2% | ±0.9% |
| CACTUS-DINO | 69.4% | 71.0% | 77.0% | 80.2% | **74.4%** | **77.7%** | 65.3% | 65.4% |
| | ±0.2% | ±0.1% | ±0.3% | ±1.0% | ± **0.3%** | ± **0.3%** | ±0.3% | ±1.6% |
| **DRESS** | **73.8%** | **76.6%** | **77.4%** | **81.6%** | 68.3% | 70.8% | **66.6%** | **68.3%** |
| | ± **0.1%** | ± **0.4%** | ± **0.1%** | ± **0.4%** | ±0.5% | ±0.0% | ± **0.6%** | ± **1.1%** |

## 5.2 Experimental Results on Real-World Datasets

We report few-shot classification accuracies on the three CelebA setups as well as the LFWA cross-domain setup from Section 4.4 in Table 2. DRESS outperforms all unsupervised methods on CelebA-Hair, excelling at capturing secondary features (i.e. hair features) beyond primary facial attributes. It also ranks first on CelebA-Primary, slightly ahead of CACTUS-DINO. We note that DINOv2, as a state-of-the-art high capacity vision encoder, is expected to capture information from the main objects (i.e. the faces), so CACTUS performs well here. On CelebA-Random, DRESS falls behind CACTUS-DINO but remains superior to other

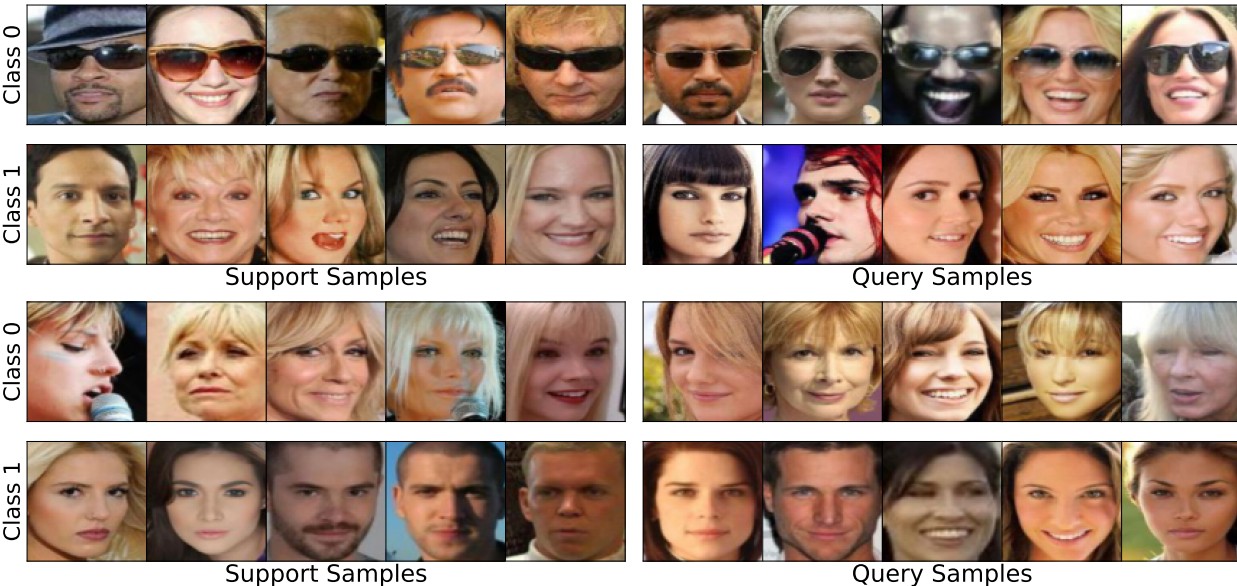

Figure 4: Two self-supervised tasks constructed by DRESS on CelebA. **Top:** Task focuses on identifying the presence of eyeglasses. **Bottom:** Task focuses on identifying the hairstyle with bangs.

Table 3: Ablation on Disentangled Representations, Latent Dimension Alignment, and Individual Dimension Clustering.

| Method | Shapes3D | Causal3D | MPI3D-Hard | CelebA-Hair | CelebA-Primary |
|---|---|---|---|---|---|
| DRESS | **93.1**% ± **0.2**% | **76.4**% ± **0.4**% | **85.0**% ± **0.5**% | **73.8**% ± **0.1**% | **77.4**% ± **0.1**% |
| DRESS w/o Disent. Repsent. | 75.3% ±0.4% | 54.0% ±0.4% | 78.8% ±0.3% | 68.9% ±0.2% | 77.3% ±0.2% |
| DRESS w/o Lat. Dim. Align. | - | - | - | 73.0% ±0.2% | 76.1% ±0.3% |
| DRESS w/o Ind. Dim. Cluster. | 80.3% ±0.8% | 76.1% ±0.2% | 66.6% ±0.5% | 72.7% ±0.4% | 74.2% ±0.3% |

baselines. This drop likely stems from the fact that disentangled representations struggle to model fine details like *bags under eyes* and *bushy eyebrows*. We confirm this by visualizing the learned disentangled latent factors in Appendix I, which indeed shows that the latent factors fail to zoom into the above-mentioned fine details within the faces. We emphasize again that despite the practical and imperfect disentangled latent factors, DRESS outperforms other methods. As disentangling encoders continue to improve and compute higher quality latent factors, we believe the DRESS will also benefit. Supervised-Original still performs poorly, showing that labels can misguide adaptation to unseen tasks. Lastly, shown by the LFWA-Cross-Domain results, DRESS also comes first when the meta-training and meta-testing data belongs to different domains, indicating more robust and transferable representations learned by the model. We provide visualizations of two tasks constructed by DRESS on CelebA in Figure 4, with additional visualizations provided in Appendix H that show the diverse facial attributes DRESS captures for constructing tasks.

### 5.3 Ablation Studies

We present in Table 3 ablation studies on each key design decision of DRESS.

**Disentangled Representations**: We replace the disentanglement learning encoder (*i.e.*, FDAE or LSD) with DINOv2, a state-of-the-art vision encoder that offers high-capacity representations but does not explicitly target disentanglement. After extracting representations from DINOv2, we follow the remaining steps of DRESS. Without disentanglement, the latent dimensions do not correspond to integral features within the data, leading to less meaningful self-supervised tasks and degradation of meta-learning capability. These

Table 4: Task Diversity Score on each dataset. The diversity score is presented here as $1-$IoU, with a higher score indicating greater task diversity.

| Method | SmallNORB | Shapes3D | Causal3D | MPI3D-Hard | CelebA-Hair |
|---|---|---|---|---|---|
| Supervised-Original | 0.95 $\pm$0.02 | 0.97 $\pm$0.01 | 0.99 $\pm$0.00 | 0.95 $\pm$0.01 | 0.89 $\pm$0.00 |
| Supervised-All | 0.98 $\pm$0.00 | 0.99 $\pm$0.00 | 0.99 $\pm$0.00 | 0.99 $\pm$0.00 | 0.90 $\pm$0.01 |
| Supervised-Oracle | 0.99 $\pm$0.00 | 0.99 $\pm$0.00 | 0.98 $\pm$0.01 | 0.98 $\pm$0.01 | 0.85 $\pm$0.01 |
| CACTUS-DC | 0.71 $\pm$0.01 | 0.81 $\pm$0.01 | **0.88**$\pm$**0.00** | 0.79 $\pm$0.00 | 0.93 $\pm$0.00 |
| CACTUS-DINO | 0.57 $\pm$0.00 | 0.61 $\pm$0.00 | 0.64 $\pm$0.00 | 0.57 $\pm$0.00 | 0.73 $\pm$0.00 |
| **DRESS** | **0.74**$\pm$**0.01** | **0.90**$\pm$**0.01** | 0.74 $\pm$0.00 | **0.91**$\pm$**0.00** | **0.98**$\pm$ **0.00** |

ablations indicate that competitive few-shot performance is driven by leveraging disentangled representations to construct diverse tasks, as in DRESS, rather than by the choice of any specific encoder.

**Latent Dimension Alignment**: As per Section 5.1, when using FDAE as the encoder, there is no explicit alignment required. Thus, this ablation study focuses on DRESS with the LSD encoder. For the ablation, we skip the process of clustering the attention masks and re-ordering the attention slots. Without alignment, the same feature dimension may express different semantic concepts on different datapoints. Small but consistent performance degradation is observed for both CelebA setups.

**Clustering within each Disentangled Latent Dimension**: Instead of performing independent clustering on each dimension, we directly cluster the entire latent space to generate the partitions. We then apply the final stage of DRESS to create self-supervised tasks from the obtained partitions. When clustering all dimensions together, the generated tasks will no longer cleanly distinguish separate factors of variation in the data. The benefits of clustering within individual latent dimensions are evident by the performance margins especially in Shapes3D, MPI3D-Hard, and CelebA-Primary.

## 5.4 Quantitative Results on Task Diversity

We compute the class-partition based task diversity, as proposed in Section 3.6, for tasks created by DRESS and applicable baselines. The task diversity scores presented are $1-$IoU, with IoU being the average value across sampled class partition pairs from each method (more details on computing the scores are provided within Algorithm 2 in Appendix J). Table 4 shows that DRESS produces more diverse tasks than CACTUS, which uses clustering in an embedding space, without the disentanglement and alignment that DRESS utilizes. For supervised meta-learning methods, the task diversity scores are computed on the partitions constructed as in Section 4.4. They serve as upper bounds on the task diversity from each dataset, as they leverage the knowledge of the ground-truth attributes or factors of variations.

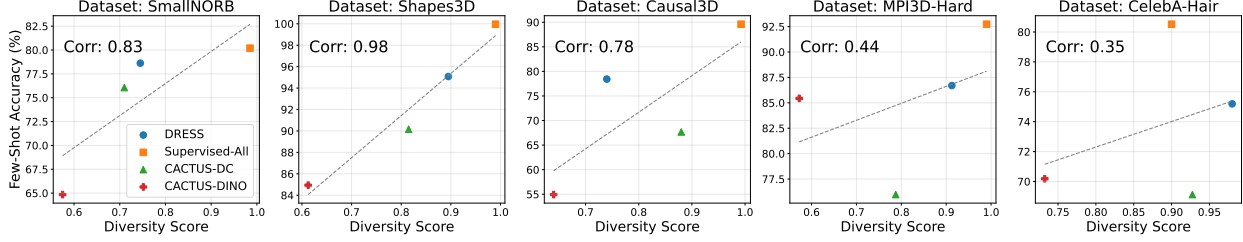

Figure 5: Correlation plots between the proposed diversity scores and the few-shot adaptation accuracies from each method on each dataset.

In Figure 5 we visualize the correlation between our proposed task diversity scores and the few-shot adaptation performance from our experiments (for which we take the average of the classification accuracies under 5-shot and 10-shot adaptations). Note that we use only one of the three supervised baselines (Supervised-All), as both Supervised-Original and Supervised-Oracle rely on hand-selection of attributes for the meta-testing tasks. On simpler datasets with distinctive and clearly defined factors of variations (SmallNORB, Shapes3D, and Causal3D) we see strong correlation between our proposed task diversity metric and few-shot adaptation performance. However, as the datasets grow in complexity, the correlation become weaker, though remains

positive showing that the task diversity metric can still indicate better adaptation performance. Specifically, on these more complex datasets, DRESS creates the most diverse meta-training tasks while achieving the best performance compared to the CACTUS-based methods.

## 6 Conclusion

We surfaced an issue in popular few-shot learning benchmarks: tasks are not diverse enough to truly test model adaptation ability. Instead, tasks with distinct natures can serve as more informative benchmarks. We proposed a self-supervised meta-learning approach that harnesses the expressiveness of disentangled representations to construct self-supervised tasks. Our approach enables models to acquire broad knowledge on underlying factors in a dataset, and quickly adapt to unseen tasks. Experimental results validate that our approach empowers the model to adapt quickly when faced with highly diverse meta-testing tasks.

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

# A Visualization and Discussions of Meta-Learning Algorithms

## A.1 Meta-Learning on Few-Shot Learning Pipeline

We provide the visualization for the general pipeline on applying meta-learning to solve few-shot learning tasks in Figure 6.

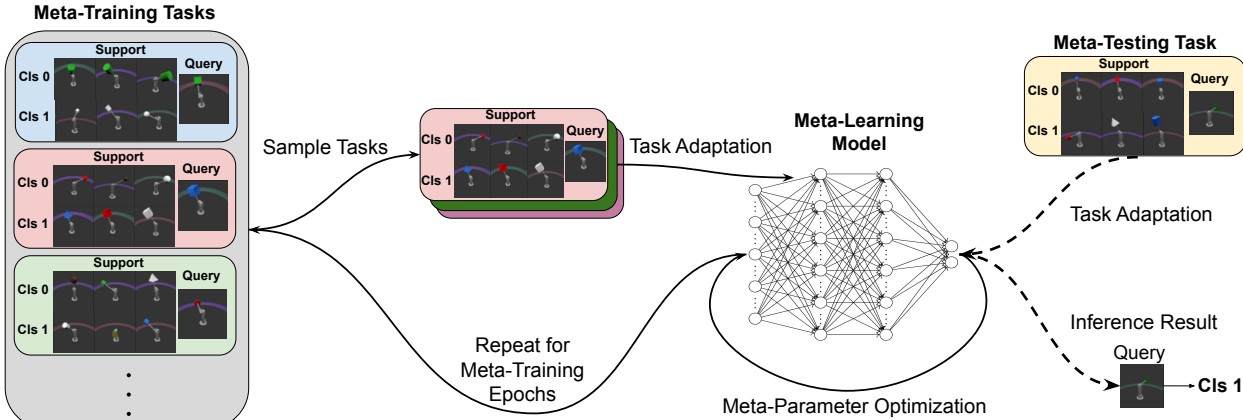

Figure 6: During the meta-training stage, the model adapts on batches of sampled tasks. The model's performance is optimized for meta-parameter optimization. After meta-training, the model can be quickly adapted to meta-testing tasks and perform few-shot inference.

## A.2 Discussions on Suitable Selection of Meta-Learning Algorithm

The majority of meta-learning algorithms can be categorized under one of the three general themes: black-box adaptation Ravi & Larochelle (2017); optimization-based adaptation, with MAML Finn et al. (2017) being the notable example; and non-parametric adaptation, with ProtoNet Snell et al. (2017) and RelationNet Sung et al. (2018) being the notable examples.

The non-parametric adaptation scheme often relies on a single pre-trained latent space, based on which the adaptation to new tasks is achieved (*e.g.*, the computation of the *prototypes* in ProtoNet, or the image embedding space in RelationNet on top of which the relation score is computed). However, as we have advocated through the design of DRESS, we need different partitions on a given dataset based on disentangled latent dimensions to correspond to different semantics or nature of diverse tasks, which is particularly important for adapting to agnostic new tasks. Therefore, while there is no fundamental incompatibility, the non-parametric adaptation scheme lacks the capacity for fully benefiting from DRESS. We also opt out of the black-box adaptation scheme for its lack of inductive bias in the learning process, due to this same reason.

Optimization-based meta-learning algorithms are suitable to combine with DRESS for learning tasks with diverse natures. This class of algorithms does not impose the assumption that the model should adapt to all the tasks based on any specific latent space, therefore allowing the model the flexibility in learning different fundamental concepts and attributes from the data, and benefiting from the comprehensive set of meta-training tasks provided by DRESS.

# B Dataset Descriptions

## B.1 SmallNORB

SmallNORB contains 48,600 images, of which we use 24,300 images for meta-training and 24,300 images for meta-testing, following the pre-defined train-test split convention on the dataset. Each image has a resolution of 96×96 pixels with a single gray-scale color channel. We simply repeat this channel three times to create

three-channel images to be compatible with all of the encoders tested (such as the pre-trained DINOv2, which expects three-channel images as inputs off-the-shelf).

The images in the dataset include 5 factors of variations, as detailed in Table 5. Note that we ignored the additional factor of *camera ID* in SmallNORB, as we exclusively take images from the first camera.

Table 5: Factors of Variation in SmallNORB

| Attribute Name | Cardinality | Constructed Tasks |
|---|---|---|
| Generic Category | 5 | Meta-Train |
| Instance ID | 5 | Meta-Train |
| Elevation Angle | 18 | Meta-Test |
| Azimuth Angle | 9 | Meta-Test |
| Lighting | 6 | Meta-Test |

### B.2 Shapes3D

Shapes3D contains 480,000 images, of which we use 400,000 images for meta-training and 50,000 images for meta-testing, following the pre-defined train-test split convention on the dataset. Each image has a resolution of 64×64 pixels with RGB color channels.

The images in the dataset include 6 factors of variations, as detailed in Table 6.

Table 6: Factors of Variation in Shapes3D

| Attribute Name | Cardinality | Constructed Tasks |
|---|---|---|
| Floor Hue | 10 | Meta-Test |
| Wall Hue | 10 | Meta-Test |
| Object Hue | 10 | Meta-Train |
| Scale | 8 | Meta-Train |
| Shape | 4 | Meta-Train |
| Orientation | 15 | Meta-Test |

### B.3 Causal3D

Causal3D contains 237,600 images, of which we use 216,000 images for meta-training and 21,600 images for meta-testing, following the pre-defined train-test split convention on the dataset. Each image has a resolution of 224×224 pixels with RGB color channels.

The images in the dataset include 7 factors of variations, as detailed in Table 7. Each of these factors are continuous values in the original form, which we have quantized to 10 levels. We emphasize that in DRESS and the competing unsupervised methods we experimented with, the models are agnostic to the quantization decision (i.e. there are 10 different values in each latent dimension that we use for creating meta-testing few-shot learning tasks). Note that the original dataset also provides labels for additional factors which we neglected in our experiments, such as rotation angles.

### B.4 MPI3D

MPI3D consists of four dataset variants. We utilize the *MPI3D_toy* dataset containing simplistic rendered images with clear color contrast. Throughout the paper, we refer to this dataset simply as MPI3D. The dataset contains 1,036,800 images, of which we use 1,000,000 images for meta-training and 30,000 images for meta-testing, following the pre-defined train-test split convention on the dataset. Each image has a resolution of 64×64 pixels with RGB color channels.

Table 7: Factors of Variation in Causal3D

| Attribute Name | Cardinality | Constructed Tasks |
|---|---|---|
| X Position | 10 | Meta-Train |
| Y Position | 10 | Meta-Train |
| Z Position | 10 | Meta-Train |
| Object Color | 10 | Meta-Train |
| Ground Color | 10 | Meta-Test |
| Spotlight Position | 10 | Meta-Test |
| Spotlight Color | 10 | Meta-Test |

The images in the dataset include 7 factors of variations, as detailed in Table 8. We note that for the two factors *horizontal axis* and *vertical axis*, denoting the robot arm's angular position, the ground truth labels for each are based on a 40-interval partition of the entire 180-degree anglular range, leading to a mere 4.5-degree maximum angle difference for two different factor values. In our experiments, we re-group the partitions into 10 intervals for each of the two axes, leading to an 18-degree maximum angle difference between two factor values.

Table 8: Factors of Variation in MPI3D under each Task Setup

| Attribute Name | Cardinality | MPI3D-Easy Task Setup | MPI3D-Hard Task Setup |
|---|---|---|---|
| Object Color | 6 | Meta-Train | Meta-Train |
| Object Shape | 6 | Meta-Train | Meta-Train |
| Object Size | 2 | Meta-Train | Meta-Train |
| Camera Height | 3 | Not Used | Meta-Test |
| Background Color | 3 | Not Used | Meta-Test |
| Horizontal Axis | 40 | Meta-Test | Not Used |
| Vertical Axis | 40 | Meta-Test | Not Used |

## B.5    CelebA

CelebA consists of 202,599 images of celebrity faces, of which we follow the conventional split and use 162,770 images for meta-training and the remaining images for meta-testing. Each image has a resolution of 178×218 pixels with RGB color channels. We conduct a cropping around the face regions in these images before feeding them into each model, for both meta-training and meta-testing.

The images in the dataset include 40 binary factors of variations. Instead of listing out all these 40 factors, in Table 9, we only list the binary attributes reserved for meta-testing few-shot learning tasks under each attribute split setup. The remaining attributes were used for constructing meta-training tasks exclusively for supervised baselines.

## B.6    LFWA

LFWA (Labeled Faces in the Wild with Attributes) consists of 13,233 images of faces of public figures, of which we use 2,000 randomly sampled images for meta-testing. Each image has a resolution of 250×250 pixels with RGB color channels. We conduct a center cropping in these images before feeding them into each model.

The images in the dataset include 73 factors of variations, with values generated using a model from the original paper in which the dataset is presented. The original values for these factors (or attributes) are float numbers. We convert them into binary values through simple thresholding. In Table 10, we list the binary attributes reserved for meta-testing few-shot learning tasks.

Table 9: Factors of Variation in CelebA under each Task Setup

| Task Setup | Attribute Name |
|---|---|
| CelebA-Hair | Bangs |
| | Black Hair |
| | Blond Hair |
| | Brown Hair |
| | Gray Hair |
| | Receding Hairline |
| | Straight Hair |
| | Wavy Hair |
| CelebA-Primary | Bald |
| | Big Lips |
| | Big Nose |
| | Blond Hair |
| | Eyeglasses |
| | Pale Skin |
| | Straight Hair |
| | Wearing Hat |
| CelebA-Random | 5 o'Clock Shadow |
| | Bags under Eyes |
| | Bald |
| | Blurry |
| | Bushy Eyebrows |
| | Double Chin |
| | Goatee |
| | Mouth Slightly Open |

Table 10: Factors of Variation in LFWA-Cross-Domain Task Setup

| Task Setup | Attribute Name |
|---|---|
| LFWA-Transfer | Big Nose |
| | Bangs |
| | Blond Hair |
| | White |
| | Sunglasses |
| | Rosy Cheek |
| | Mouth Closed |
| | Pale Skin |

## C   Detailed Setups for Pre-training and Fine-tuning

For pre-training, we use an encoder backbone that shares the same architecture as the ResNet-18 He et al. (2016) backbone used for FDAE. After pre-training, a trainable linear layer is attached on top of the encoder for the adaptation process on evaluation tasks. The encoder is frozen throughout the adaptation process. We include the details for this approach in Table 11. Note that we do not use a supervised loss in pre-training in order to avoid the encoder focusing only on tasks that are irrelevant to the meta-evaluation tasks, as we have discussed in Section 3.

Regarding the number of epochs for pre-training, in the pre-training procedure the entire set of meta-training image inputs are fed to the encoder (i.e. 400,000 images for Shapes3D; and 1,000,000 images for MPI3D). Therefore, with 10 epochs over the entire meta-training dataset, the number of forward-backward computations for optimizing the encoder already surpasses the models trained with the meta-learning methods.

Table 11: Pre-Training and Fine-Tuning Setup

| Setting | Value |
|---|---|
| Pre-Training Epochs | 10 |
| Tasks in Meta-Evaluation | 1000 |
| Gradient Descent Steps in Adaptation | 5 |

## D   Additional Setup Details for Meta-Learning Methods

In this section, we further provide more details on the implementation of DRESS as well as meta-learning baselines.

Firstly, for DRESS, the supervised meta-learning baselines, as well as the two CACTUS baselines, we use MAML Finn et al. (2017) as the meta-optimization engine, with a convolutional neural network (CNN) of identical specification as the base learner, for fair comparisons between the methods. The few-shot direct adaptation baseline also uses a CNN of the same specification. For the remaining baselines, we follow the design details as in the original papers.

The number of visual concepts (or attention slots) that we adopt to for each dataset is not the same as the number of the ground-truth factors of variations. For example, factors such as the orientation angle of the object, the lighting condition, or the camera height do not necessarily correspond to individual visual concepts, but instead are reflected by the relations among multiple visual concepts. Therefore, the number of visual concepts that we use in DRESS for each dataset is estimated based on the nature of the image composition in each dataset. We provide the values we used for the experiments in Table 12. We note that in early explorations, the few-shot adaptation results were not sensitive to mild changes on these values. These values are also not extensively optimized.

Table 12: Number of Visual Concepts or Attention Slots on Each Dataset

| Dataset | SmallNORB | Shapes3D | Causal3D | MPI3D | CelebA |
|---|---|---|---|---|---|
| Number of Visual Concepts | 8 | 6 | 8 | 7 | 12 |

We summarize in Table 13 and Table 14 respectively the hyper-parameter values of DRESS as well as the meta-learning baselines CACTUS-DeepCluster and CACTUS-DINOv2. The selections of the hyper-parameter values are largely based on the specifications from the original paper Hsu et al. (2019) (while the number of clusters over each latent space is originally 500, through our experiments, we find that using 300 clusters leads to no noticeable performance change over various datasets). For the DINOv2 encoder, we use the ViT-S/14 distilled version with registers. We note that for the DeepCluster encoder, PCA is applied on its output to reach the number of latent dimensions as listed.

Table 13: Task Construction Setup for DRESS

| Setting | DRESS-FDAE | DRESS-LSD |
|---|---|---|
| Reduced Number of Components per Latent Dimension | 40 | - |
| Clusters in Each Latent Dimension | 200 | 200 |

In Table 15, we provide meta-training and meta-testing hyper-pamameters for DRESS and two meta-learning baselines, CACTUS-DC and CACTUS-DINOv2.

Table 14: Task Construction Setup for CACTUS-based Baselines

| Setting | CACTUS-DC | CACTUS-DINOv2 |
|---|---|---|
| Latent Dimensions | 256 | 384 |
| Randomly Scaled Latent Spaces | 50 | 50 |
| Clusters Over Each Latent Space | 300 | 300 |

Table 15: Few-Shot Learning Setup for All Meta-Learning Methods

| Setting | Value |
|---|---|
| Tasks per Meta-Training Epoch | 8 |
| Meta-Training Epochs | 30,000 |
| Tasks in Meta-Evaluation | 1,000 |
| Gradient Descent Steps in Task Adaptation | 5 |
| Adaptation Step Learning Rate | 0.05 |
| Meta-Optimization Step Learning Rate | 0.001 |

## E  Computation Details

All the experiments include training and evaluating models on each dataset are conducted on one or two Nvidia RTX 6000 Ada Generation GPUs, each with 48GB memory, under the standard Ubuntu OS (Ubuntu 24.04.1 LTS). Our code implementation is based on the PyTorch library.

In terms of the computational cost of each method, the cost of performing few-shot adaptation is negligible for every method, therefore the computational cost is dominated by the training stage. For both DRESS and CACTUS, the training process involves two steps: training the encoder and training the meta-learner model. For both Meta-GMVAE and PsCo, the meta-training process is coupled with training the encoder by the algorithm design. For meta-training on the CelebA dataset, we report the computation time in Table 16.

Table 16: Training Time for each Encoder on CelebA

| Encoder | LSD in DRESS | DeepCluster in CACTUS | DINOv2 in CACTUS | Meta-GMVAE | PsCo |
|---|---|---|---|---|---|
| Encoder training time | 8.5 Hours | 12.3 Hours | 3.3 Days[2] | 7.0 Hours | 21.0 Hours |
| Meta-Train with constructed tasks | 2.9 hours | 2.9 hours | 2.9 hours | | |

## F  Latent Dimension Alignment Process

As described in Section 5.1, when using the LSD encoder for DRESS, we need the explicit latent dimension aligment process. From the LSD encoder, each *visual concept* is modeled in the latent space via an attention mask and an encoding vector. When encoding multiple images with LSD, the order of the obtained latent

---

[2]As reported by the creators for the large model, DINOv2 ViT-L/14, on a large multi-GPU hardware setup. We used DINOv2 ViT-S/14, which used more computation overall as it was distilled from the large model.

representations for these visual concepts are stochastic. We use the encoding vectors of all the visual concepts from each image as the disentangled latent representations for DRESS. To properly align these encoding vectors, we perform a consistent semantic ordering for the attention masks across all input images.

Our detailed procedure is as follows: for the first batch of input images, we obtain their latent representations, including the attention masks, from the LSD encoder. We then perform K-Means clustering with a predefined number of clusters. Through our experiments, the best results are obtained when the number of clusters equals to the number of visual concepts we extract from each image. With the obtained clusters, for each image in the dataset, we obtain the cluster identities of its attention masks, and order its encoding vectors following an arbitrary but fixed order of the cluster identities. We note that there are images whose attention masks are not strictly clustered among all the clusters uniformly, i.e. there are clusters with more than one attention mask and clusters with zero attention mask. For these corner cases, we simply break the tie by distributing attention masks from larger clusters to empty clusters, such that we always end up with one attention mask in each cluster for each image before we perform the alignment.

In Figure 7, we provide visualizations on the attention mask ordering for the LSD encodings on a random set of images before and after our latent dimension alignment procedure. Evident from the visualization, after the latent dimension alignment process, the attention masks are aligned reasonably well across all the images with just a few misaligned regions. Therefore, under each disentangled latent dimension, the region as the focal point stays very close from image to image, ensuring that the latent representations are semantically aligned across all the images. We note that we choose this attention-mask based alignment process based on the fact that the composition is consistent across images in the dataset studied (i.e. in the CelebA dataset, all the images are centered on individual faces under the natural orientation).

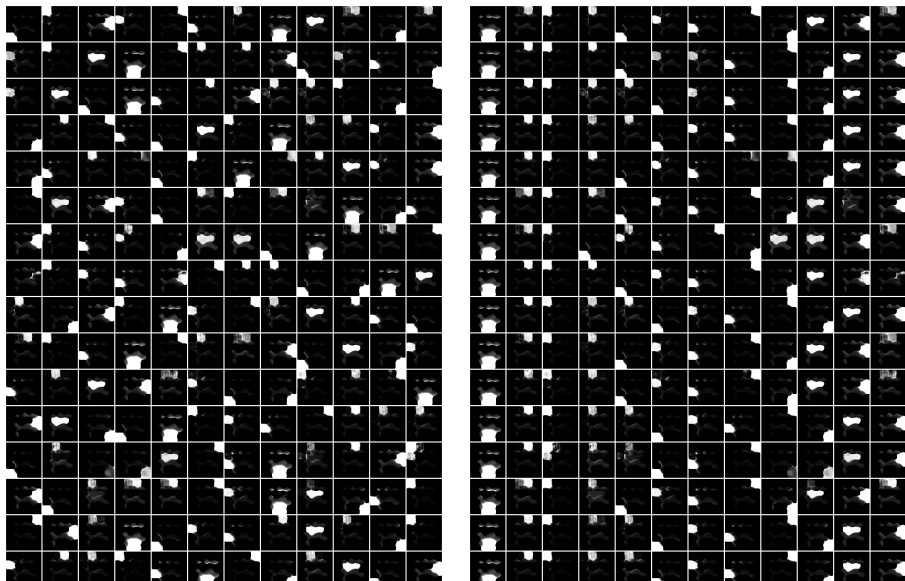

Figure 7: The ordering of attention masks from LSD encoding. In both subfigures, each row lists the ordered attention masks of an image. **Left:** Latent Dimension ordering *before* the alignment process. **Right:** Latent Dimension ordering *after* the alignment process on the same input images.

# G Experiments on Low Task-Diversity Benchmarks

To complement our analysis on the effect from the dataset-intrinsic task-diversity, we provide further experiments on the low-task-diversity benchmark. Specifically, we present the few-shot adaptation results on the Omniglot dataset in Table 17.

As shown by the results, in the low-task-diversity benchmark regime, the Pre-training and Fine-tuning approach regains its competitiveness compared to meta-learning approaches, which aligns with our earlier hypothesis regarding the comparisons between these two classes of methods. Among the meta-learning methods, CACTUS-DINO achieves the best performances. Due to the lack of independent factors of variations within the low-diversity dataset, DRESS does not show its unique advantage from exploiting the disentangled latent space, although still achieving respectable performances among all competing methods.

Table 17: Few-shot classification accuracies on the Omniglot dataset, with each trial conducted over 1000 meta-testing few-shot learning tasks.

| Method | Omniglot | |
| --- | --- | --- |
| | 5-Shot | 10-Shot |
| Supervised-Original | $99.2\%$ $\pm 0.1\%$ | $99.4\%$ $\pm 0.0\%$ |
| FSDA | $82.8\%$ $\pm 0.7\%$ | $80.8\%$ $\pm 0.4\%$ |
| PTFT | $97.7\%$ $\pm 0.3\%$ | $98.3\%$ $\pm 0.1\%$ |
| CACTUS-DC | $88.8\%$ $\pm 0.6\%$ | $90.3\%$ $\pm 0.4\%$ |
| CACTUS-DINO | $99.1\%$ $\pm 0.1\%$ | $99.4\%$ $\pm 0.1\%$ |
| **DRESS** | $93.7\%$ $\pm 0.1\%$ | $94.9\%$ $\pm 0.1\%$ |

## H  Additional Task Visualizations from DRESS

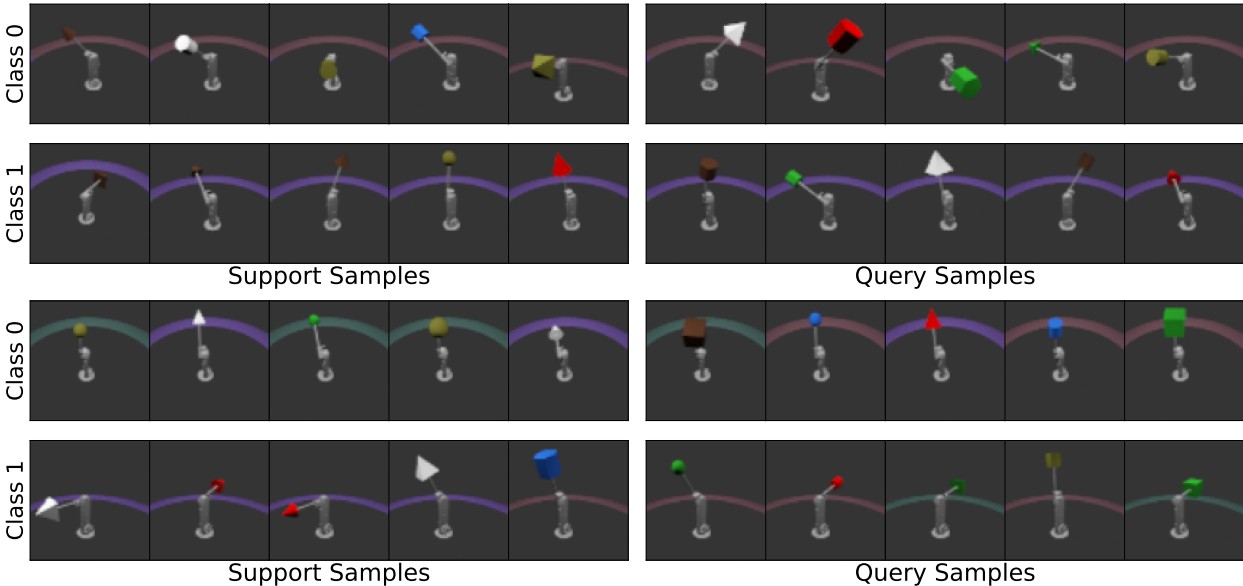

Figure 8: More self-supervised tasks constructed by DRESS on MPI3D. The top task focuses on the background color; while the bottom task focuses on the camera height.

We provide more visualizations on self-supervised few-shot learning tasks generated by DRESS on MPI3D in Figure 8, as well as tasks generated by DRESS on CelebA in Figure 9. As evidenced by these visualizations, the generated tasks have very distinctive natures, covering multiple aspects and factors of variations within the corresponding datasets. When being trained on such diversified tasks, the resulting model naturally acquires the ability to adapt well on unseen tasks, regardless of the semantics that the tasks focus on.

## I  Visualizing Learned Disentangled Latent Factors

Specifically, we provide in Figure 10 and Figure 11 the attention masks for latent factors learned by the FDAE encoder, on MPI3D and Shapes3D respectively. Furthermore, we provide in Figure 12 the attention

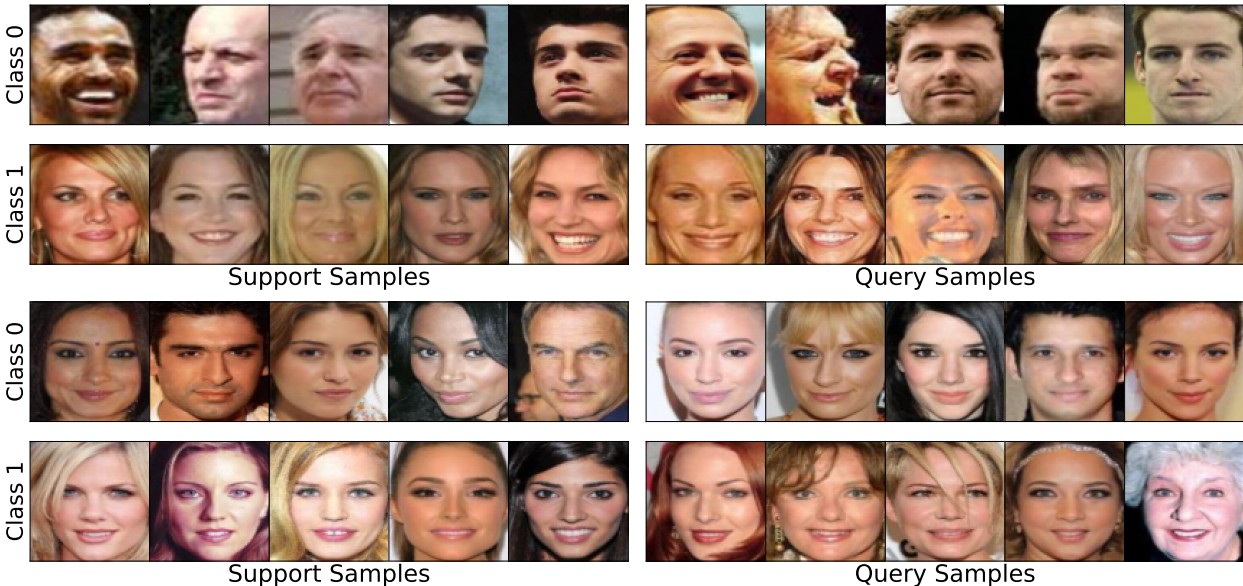

Figure 9: More self-supervised tasks constructed by DRESS on CelebA. The top task focuses on the gender of the person; while the bottom task focuses on if the person has mouth open or not.

slots learned by the LSD encoder on CelebA. Note that the factors shown in Figure 12 have not been aligned yet. The alignment procedure is elaborated above in Section F.

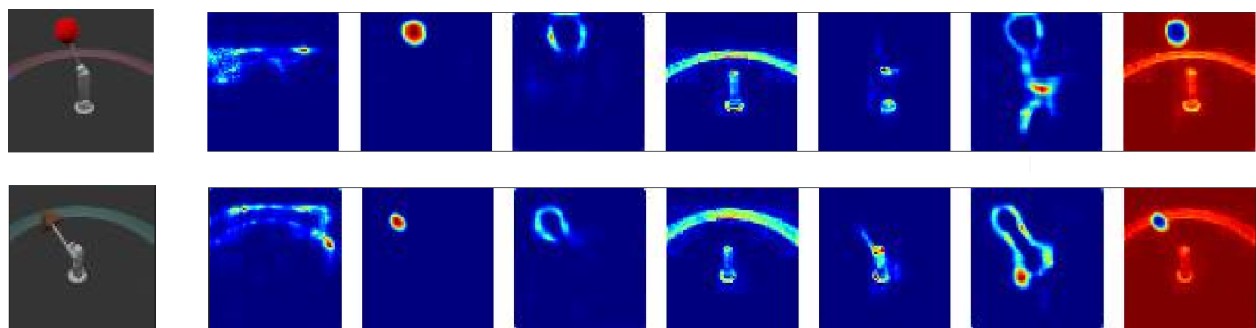

Figure 10: Visualization of attention masks of disentangled factors learned by the FDAE encoder on MPI3D Images. **Left:** Original image. **Right:** Attention masks from disentangled factors.

## J   Computation Details for Class-Partition based Task Diversity Metric

With the task diversity defined in Section 3.5, we aim to compute the intersection-over-union ratio (IoU) over pairs of tasks created by each method. Nonetheless, as we focus on the few-shot learning tasks (five-shot two-way tasks to be specific), the number of input samples on each task is very small. Therefore, if we directly take two such few-shot learning tasks, there is most likely no intersection in the samples they cover.

To address this difficulty, we instead focus on the partitions over the entire dataset. As described in Section 3 and Section 4.4, for DRESS, supervised meta-learning baselines, as well as the CACTUS-based baselines, the individual tasks are directly sampled from the dataset-level partitions. Therefore, computing the diversity metric over these partitions can give us a good proxy to the evaluation of the task diversity from each method. We now present the procedure for computing the class-partition based task diversity.

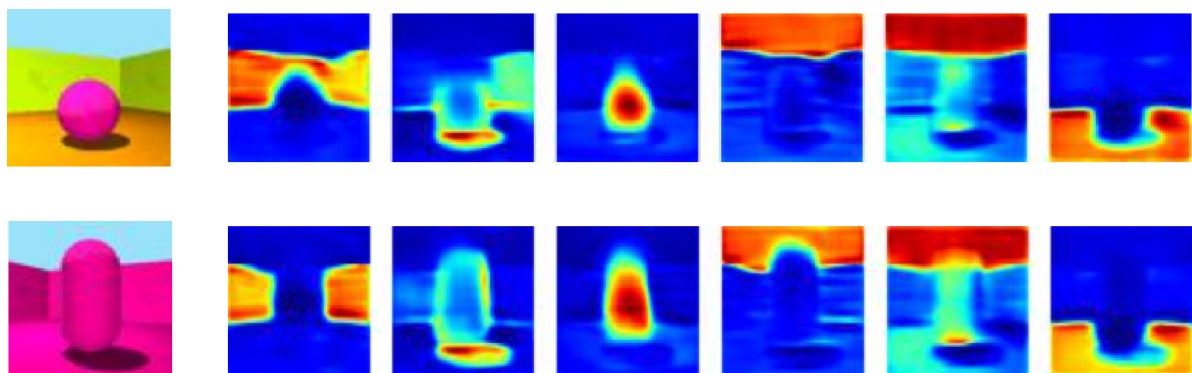

Figure 11: Visualization of Attention Masks of Disentangled Factors Learned by the FDAE Encoder on Shapes3D Images.

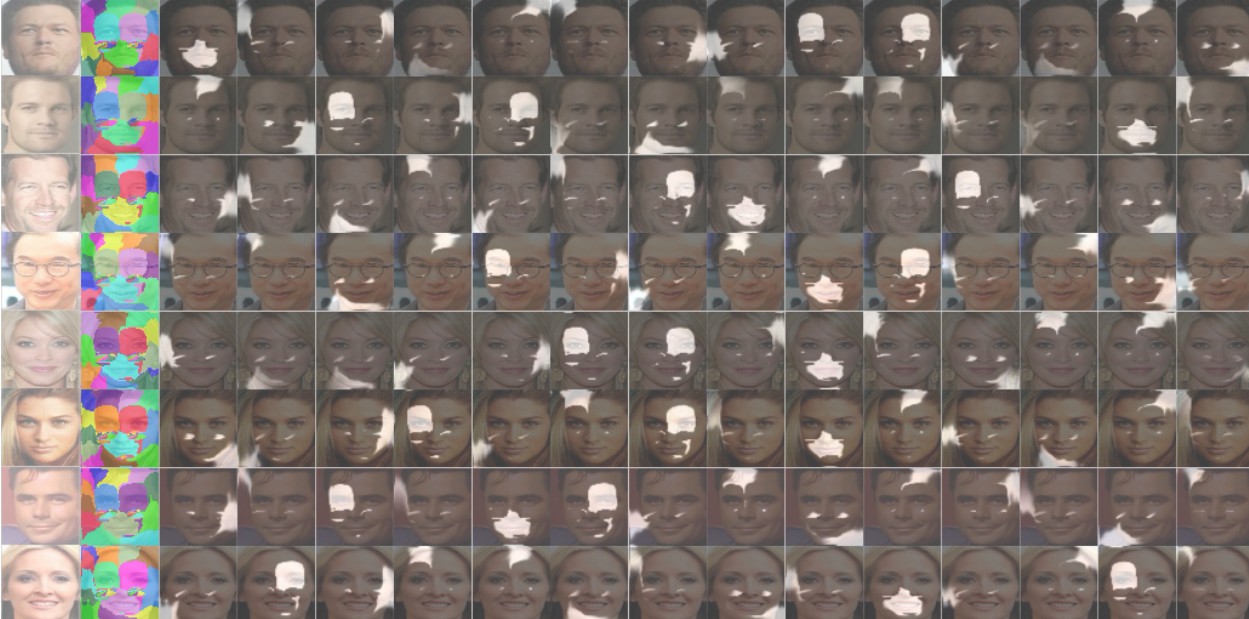

Figure 12: Visualization of Attention Slots of Disentangled Factors Learned by the LSD Encoder on CelebA Images (before alignment).

Consider two partitions on the same dataset generated by a specific meta-learning method: $\mathcal{P}^1 = \{\mathcal{P}^1_i\}_{i=1}^{K_p}$ and $\mathcal{P}^2 = \{\mathcal{P}^2_i\}_{i=1}^{K_p}$, where $\mathcal{P}^1_i$ and $\mathcal{P}^2_i$ denotes the $i$-th subset in $\mathcal{P}^1$ and $\mathcal{P}^2$ respectively, and $K_p$ is the number of subsets in each partition. Note that if one partition has fewer subsets, we can simply regard it as having extra empty subsets, such that the total number of subsets reaches $K_p$. We use these dataset partitions to replace the class partitions defined from the counterpart pair of supervised tasks, i.e. $\{\mathcal{P}^1_{c_j}\}$ and $\{\mathcal{P}^2_{c_j}\}$ as defined in Section 3.6.

We summarize our procedure for computing the values on the purposed task diversity metric in Algorithm 2. Note that instead of performing strict bipartite matching for subsets between $\mathcal{P}^1$ and $\mathcal{P}^2$, we match the subsets through a greedy process: going through the subsets one-by-one in $\mathcal{P}^1$, and find the best match from the remaining subsets in $\mathcal{P}^2$. While this greedy procedure does not strictly guarantee the perfect matches between the two partitions, it provides a decent estimates for our quantitative analysis at a manageable level of computational cost.

To produce the diversity scores as reported in Table 4, within the set of partitions that each method creates on a given dataset, we uniformly sample 30 pairs of distinct partitions. Each pair of partitions is fed as inputs to Algorithm 2 to compute the pairwise diversity score. The average of the 30 diversity scores is then reported as the expected task diversity score from each method.

---

**Algorithm 2** Task Diversity Metric Computation Procedure

---

1: **Input:** $\mathcal{P}^1 = \{\mathcal{P}^1_i\}_{i=1}^{K_p}$, $\mathcal{P}^2 = \{\mathcal{P}^2_i\}_{i=1}^{K_p}$
2: idx_list $\leftarrow [1, 2, \ldots, K_p]$
3: IoU_list $\leftarrow \emptyset$
4: **for** $i \leftarrow 1$ to $K_p$ **do**
5:     idx_matched $\leftarrow 0$
6:     highest_IoU $\leftarrow 0$
7:     **for** $j \in$ idx_list **do**
8:         IoU $= \frac{|\mathcal{P}^1_i \cap \mathcal{P}^2_j|}{|\mathcal{P}^1_i \cup \mathcal{P}^2_j|}$
9:         **if** IoU $>$ highest_IoU **then**
10:            idx_matched $\leftarrow j$
11:            highest_IoU $\leftarrow$ IoU
12:         **end if**
13:     **end for**
14:     IoU_list.append(highest_IoU)
15:     **if** idx_matched $> 0$ **then**
16:         idx_list.pop(idx_matched)
17:     **end if**
18: **end for**
19: avg_IoU_score $\leftarrow$ avg(IoU_list)
20: diversity_score $\leftarrow 1 -$ avg_IoU_score
21: **Output:** diversity_score

---

