# OpenReview forum: "DRESS: Disentangled Representation-based Self-Supervised Meta-Learning for Diverse Tasks"
_TMLR — Rejected by TMLR_

### Review · Reviewer_f236 · 2026-03-09

**Summary Of Contributions:**

The manuscript proposes DRESS, a task-agnostic self-supervised meta-learning approach designed to improve model adaptation on highly diversified few-shot learning tasks. The authors hypothesize that meta-learning struggles to outperform simple pre-training and fine-tuning (PTFT) baselines on standard benchmarks because those benchmarks lack "task diversity." To address this, DRESS relies on a multi-stage pipeline: it extracts disentangled latent representations using a pre-trained encoder (e.g., FDAE, LSD), explicitly aligns these semantic dimensions across images, applies independent K-Means clustering to each latent dimension to generate pseudo-classes, and constructs self-supervised few-shot tasks from these partitions. Finally, a meta-learner (e.g., MAML) is trained on these generated tasks. The method is evaluated on several attribute-centric datasets like SmallNORB, Shapes3D, and CelebA.

**Additional Comments:**

**Strengths**
*   **Interesting Problem Space:** The observation that standard few-shot benchmarks (like miniImageNet or CIFAR-FS) are fundamentally homogeneous (i.e., they only evaluate main object classification) is a valid and interesting critique of current evaluation protocols.
*   **Extensive Empirical Validation:** The authors provide thorough experiments on multiple datasets with controlled factors of variation (SmallNORB, Shapes3D, Causal3D, MPI3D) and realistic datasets (CelebA), which are well-suited for testing disentangled representations.

**Weaknesses**

Despite the interesting premise, the paper suffers from foundational flaws in its motivation, a rigid and outdated methodological design, and an unconvincing theoretical explanation of the core problem.

**1. Unproven Motivation Regarding Pre-Training vs. Meta-Learning**
The central motivation of the paper hinges on the claim that simple pre-training and fine-tuning (PTFT) universally surpasses meta-learning. This is a highly contested, context-dependent, and largely unproven overgeneralization within the community. While some recent studies have indeed highlighted that well-tuned PTFT baselines are highly competitive on specific image classification benchmarks, meta-learning algorithms inherently maintain significant advantages in continuous adaptation, structured low-data regimes, and dynamic environments. Framing the entire contribution around "rescuing" meta-learning from PTFT is a weak foundation, especially since the empirical equivalence of the two is highly dependent on backbone capacity and specific training regimens.

**2. Unconvincing Explanation of the "Task Diversity" Problem**
The authors attribute the purported shortcomings of meta-learning entirely to a "lack of task diversity," but the explanation and formalization of this problem are highly unconvincing. To quantify task diversity, the authors introduce a metric based solely on the class-partition overlap (1 - IoU) directly in the input space. This metric is excessively simplistic; it merely measures label distribution mismatch rather than true semantic, structural, or functional diversity between tasks. Consequently, treating this basic partition misalignment as the definitive measure of "task diversity" fails to capture the complex, multifaceted nature of why meta-learning models generalize well or poorly.

**3. Lack of Methodological Novelty and Reliance on Heuristics**
The methodological design of DRESS leans heavily on composing existing off-the-shelf methods (e.g., FDAE, LSD, K-Means, and MAML) rather than introducing a fundamentally novel deep learning architecture or objective. The pipeline—extracting representations, explicitly enforcing alignment, and applying independent K-Means clustering on individual 1D semantic dimensions—involves significant hand-designed features and rigid heuristics. Instead of allowing the neural network to organically discover complex, non-linear correlations and task relationships in an end-to-end fashion, DRESS artificially bottlenecks the learning process into isolated, engineered dimensions. This heavy reliance on feature engineering and the mere composition of existing algorithms severely limits the technical novelty of the contribution.

**4. Missing Crucial Literature on Deep Meta-Learning Generalizability**
Because the authors resort to a heuristic, clustering-based pipeline to generate diverse tasks, they overlook much more natural, end-to-end "deep learning" approaches designed to improve meta-learning generalizability. Integrating contrastive objectives directly into the meta-learning loop has proven to be a highly effective way to encourage robust, generalizable task representations without requiring rigid disentanglement or hand-crafted clustering rules. The authors fail to acknowledge or compare their work against recent advances in this area. E.g., [1][2], which introduces a natively deep, contrastive approach to enhancing the generalizability of meta-learning frameworks.

**5. Missing Existing Benchmarks**
There are many exiting and widely-used meta-learning benchmarks, e.g., Meta-dataset which contains multi-domains matching with this paper's motivation, and others like miniImagenet. But the authors have only evaluated on the new benchmark proposed by themselves. None of these benchmarks are evaluated.

[1] Learning to learn with contrastive meta-objective. Nips2025

[2] Robust Fast Adaptation from Adversarially Explicit Task Distribution Generation. KDD2025

**Audience:**

Yes

**Audience Explanation:**

This manuscript provide a method trying to enhance generalizability of MAML for few-shot classification.

**Claims And Evidence:**

No

**Claims Explanation:**

The claimed contribution 1 "We identify the lack of task diversity in few-shot learning benchmarks, explaining why pre-training and fine-tuning can seem to outperform meta-learning." is not supported by convincing explanation nor evidence.

**Requested Changes:**

Please refer to Additional Comments

---

> ### Author Response · Authors · 2026-03-24
> **Responses to Weakness 1 and 2 by the Reviewer**
>
> Sincere thanks to the reviewer's evaluation and comments. Please find below our responses to each point from the Weaknesses commented from the reviewer. Due to the length restriction of each reply window, we present in this window our responses to the first two issues commented by the reviewer.
>
> **1. Unproven Motivation Regarding Pre-Training vs. Meta-Learning:** We share the view with the reviewer of the importance on having the proper scope for motivation and claims, which is why we start the paper by introducing the few-shot learning as the specific problem setting focused by this paper, instead of other problem settings such as continuous adaptation. Regarding few-shot learning specifically, a line of work in the literature has indeed shown the competitiveness of Pre-training and Fine-tuning (PTFT) compared to meta-learning [1-4]. The experiments and findings from these works span across a wide range of datasets over various backbone model architectures, all under the scope of few-shot learning. Some of the results are from datasets beyond the image domain (e.g. [4]). To emphasize our idea on the task diversity, we selected well-established datasets with quality ground-truth labels across multiple factors of variations. These datasets, widely studied in the fields of representation learning and generative modeling, are predominately in the image domain. Essentially, while we agree with the reviewer that it is not a simple comparison to be drawn between meta-learning and PTFT, the evidences that PTFT is highly competitive to or even surpassing meta-learning are well documented in the literature under the specific setting of few-shot learning.
>
> **2. Unconvincing Explanation of the "Task Diversity" Problem:** We understand that it might appear concerning the way we define our task diversity metric, directly on the input space using class partitions. In fact, we have explained the rational for designing this metric directly on the input space, at the beginning of **Section 3.6**:
> > In DRESS, different encoders with different embedding spaces could be used to construct tasks. Correspondingly, we advocate for a task diversity metric that is not tied to any specific embedding space, but is directly linked to the original input space, unlike metrics such as Task2Vec.
>
> We also acknowledged that our metric does not account for information such as class semantic at the end of this section:
> > We note that during the step of relabeling the classes, the semantic information
> of the classes in each task is lost. Therefore, the proposed metric only quantifies task diversity from the
> function mapping perspective.
>
> However, having diversified function mappings during training, even if they are potentially semantic-agnostic, has been shown to improve a model's adaptation capacity in self-supervised learning [5]. Essentially, to learn each function mapping, the model has to be able to infer the underlying semantic information and the properties of the function mapping.
>
> Lastly, we do not claim that our metric can fully explain the generalization result from a class of approaches. As the reviewer rightfully noted, justifying the *"complex, multifaceted nature of why meta-learning models generalize well or poorly"* is indeed a problem that is highly challenging for any quantitative metric. We introduce this metric to provide quantitative support, instead of *"the definitive measure"*, for our discussion on the task diversity in meta-training tasks.
>
> As we have shown in **Section 5.4**, there is indeed general positive correlation observed between the proposed metric and the adaptation performance. Nonetheless, we have acknowledged in the paper that the correlation can be weak, indicating that the proposed metric is not fully justifying the entire dynamics of the adaptation process. Essentially, throughout the paper, we present our metric as a more suitable way to quantify the task diversity compared to existing metrics such as **Task2Vec**, due to the reason explained in the first quote from our paper above.
>
> ---
> [1] Rethinking Few-Shot Image Classification: a Good Embedding Is All You Need? (ECCV 2020)
>
> [2] A unified few-shot classification benchmark to compare transfer and meta learning approaches. (NeurIPS 2021)
>
> [3] Partial is better than all: Revisiting fine-tuning strategy for few-shot learning. (AAAI 2021)
>
> [4] Omni-training: Bridging pre-training and meta-training for few-shot learning. (TPAMI 2023)
>
> [5] Self-supervised representation learning from random data projectors. (ICLR 2024)

---

> ### Author Response · Authors · 2026-03-25
> **Responses to Weakness 3, 4 and 5 by the Reviewer**
>
> **3. Lack of Methodological Novelty and Reliance on Heuristics:** We organize our responses into two aspects: regarding the novelty concern; and regarding the engineering concern.
>
>   * 3.1. Regarding the novelty concern: We acknowledge the reviewer's point that DRESS does not introduce fundamentally novel deep learning architecture. Nonetheless, a large portion of machine learning researches have made notable contributions without necessarily proposing a novel architecture. Instead of proposing a novel learning architecture, the novelty of our paper is rooted from a self-contained logic chain starting from emphasising the importance of task diversity in the few-shot learning setting, to proposing a meta-learning method that actively exploits high task diversity for effective training. Essentially, we bring into attention an important issue (the effect of task diversity in the fast adaptation performance) in the fields of few-shot learning and meta learning, which has been largely overlooked previously, in both the method design and in benchmark preparation. We believe this paper has provided a comprehensive treatment to this specific topic.
>
>   * 3.2. Regarding the engineering concern: For the stages of DRESS, from the feature learning, to feature alignment, and to the task construction, the learning process has been exclusively unsupervised or self-supervised. We do not inject any hand crafted features, or impose any human guidance throughout the process. The only aspect within DRESS that is human engineered is the selection of hyper-parameters (i.e. how many dimensions or how many clusters). Such selections of hyper-parameters are rather very common in researches especially for literature related to unsupervised or self-supervised learning.
>
> **4. Missing Crucial Literature on Deep Meta-Learning Generalizability:**  We thank the reviewer for the pointers to two great works in meta-learning. Nonetheless, each of these works differ from the problem setting we are studying to a certain degree. For [1], supervised signals are assumed to be available during the meta-training stage, allowing for known task identities to be used as the additional signal to perform contrastive learning in the model weight space. In our case, as we study self-supervised meta-learning, we only assume the availability of the raw dataset during the meta-training stage, with tasks constructed from scratch in the self-supervised manner. However, we do note on the potential for combining [1] and DRESS in the learning process: after tasks with distinct identities and natures are constructed from DRESS, [1] can be utilized for contrastive learning from these self-supervised tasks. This direction for sure would be an exciting route for future exploration. For [2], it is focusing on the continuous adaptation where the meta-testing tasks would drift out-of-distribution compared to the tasks from the meta-training stage. Meanwhile, we motivate DRESS to create diversified enough tasks by covering all the independent factors of variation within the underlying data distribution, such that any task in the meta-testing stage would still be in-distribution to the wide range of self-supervised tasks created by DRESS, allowing the model for fast adaptation.
>
> Many thanks to the reviewer's comment and pointers, although the two papers referenced are not directly comparable to the methods we experimented, in the revised paper, we have included them to further strengthen the literature discussion.
>
> **5. Missing Existing Benchmarks:** The benchmarks we used are not proposed by ourselves, but have been largely studied in representation learning or generative modelling. We introduce them for studying meta-learning approaches over the existing meta-learning benchmarks, due to the exact reason that we have advocated throughout the paper. We quote our reasoning from the introduction section of our paper:
> > For instance, in canonical few-shot learning datasets such as Omniglot (Lake et al.,2011), miniImageNet (Vinyals et al., 2016), and CIFAR-FS (Bertinetto et al., 2019), the distinct tasks differ solely in that their targets belong to non-overlapping sets of object classes. In essence, these tasks all share the same nature: main object classification. Hence, there is one degenerate strategy for solving all these tasks simultaneously: compare the main object in the query image to the main objects in the few-shot
> support images, and assign the class label based on similarity to support images.
>
> Nonetheless, we agree with the author that it is important to also study the performances on the existing meta-learning benchmarks. In the revised paper, we have introduced the experiment results for **Omniglot** (and **mini-ImageNet**, for which we are still running the experiments currently, and will add in the results as soon as the results are fully collected). We thank the reviewer again for raising this point.

---

### Review · Reviewer_DHBP · 2026-03-15

**Summary Of Contributions:**

This paper contributes to the study of meta-learning with an emphasis on disentangled representations. The first contribution is a hypothesis as to why pre-training + fine-tuning often outperform meta-learning approaches in the literature: a lack of diversity in the meta-learning tasks of popular academic benchmarks. The second contribution is a set of benchmarks of increased diversity. The third, and main, contribution is DRESS: a meta-learning algorithm that makes use of disentangled representations to create a diverse set of meta-training tasks. The final contribution is an input-space task class-partition diversity metric

### Strengths
* The paper clearly identifies a limitation in the benchmarking of meta-learning algorithms in the literature.
* The paper proposes a simple to understand and generic algorithm for improving the performance of meta-learning in highly task diverse settings.
* The experimental evidence is generally clear, accurate and supportive of the paper's claims.

### Weaknesses
* While the paper thoroughly explores the high-diversity setting, it neglects the low-diversity setting (i.e., there are no experiments on Omniglot, CIFAR-FS, etc.). Results for these datasets would help to validate that the hypotheses of the paper are correct. One would expect to see that DRESS doesn't provide the strongest performance in these settings. Similarly, the proposed diversity metric should be low for these datasets. It would be great for all of the main results to be extended to these datasets.
* In this paper, DRESS is applied on top of MAML. However, results for vanilla MAML are not included. This makes it difficult to quantify the impact of DRESS.

**Audience:**

Yes

**Audience Explanation:**

The paper should be of interest to any meta-learning researcher or practitioner as it highlights a fundamental issue with benchmarking in this field. Furthermore, the paper should also be of interest to the disentangled representation community and the broader representation learning communities due to its clear link between meta-learning tasks and disentangled representations.

Furthermore, the proposed algorithm, DRESS, is agnostic to the specific meta-learning and disentanglement algorithms which makes it applicable in a wide range of settings.

**Claims And Evidence:**

No

**Claims Explanation:**

The qualitative and quantitative results are generally clear, accurate and supportive of the paper's narrative and claims. However, as mentioned above there are a few benchmarks and a baseline that are missing. I believe that these are small issues that can easily be resolved.

**Requested Changes:**

1. [Critical] please extend table 1, 3, and 4, as well as figure 5 with the omniglot and CIFAR-FA (or miniImageNet) datasets.
2. [Critical] please extend table 3 with MAML.

---

> ### Author Response · Authors · 2026-03-24
> **Additional Experiments Introduced & Establishing the presence of MAML in the paper**
>
> Sincere thanks to the reviewer's evaluation and comments. Please find below our responses to each of the weaknesses and requested changes.
>
> **Weakness 1:** We agree with the reviewer's suggestion. In the revised paper submitted, under **Appendix G**, we have included additional experiments on the low-diversity datasets including Omniglot (and mini-Imagenet, for which we are still finishing up experiments and will include the results as soon as they are available). Just like the reviewer expected, at the low-diversity regime, pre-training and fine-tuning catches up with meta-learning approaches. Also, without multiple independent factors of variations on these low-diversity datasets, DRESS does not gain explicit advantages from disentangled representations or diversified task creation like it does in high-diversity dataset regime. Instead, the raw encoder capacity emerges as the decider (as shown by CACTUS-DINO thanks to the state-of-art vision encoder DinoV2). Nonetheless, DRESS does not explicitly suffer from the change of task-diversity regime either, and still maintain respectable performance among the methods.
>
> **Weakness 2:** Actually, MAML is exactly represented by the **Supervised-Original** baseline in our paper. Furthermore, by varying the degrees of relevancy of the ground-truth information available between meta-training and meta-testing stage, we further proposed two variants on top of MAML: **Supervised-All**, and **Supervised-Oracle**. Detailed elaboration on the definitions of these baselines are provided in **Section 4.3**. We propose in this paper these three supervised baselines, as they collectively not only represent the baseline when ground-truth information is available, but also reveal the importance of the task diversity in studying few-shot learning, by varying how close the meta-testing tasks align with the meta-training tasks through these task-defining ground-truth labels. Lastly, we also discussed on the rational of selecting MAML over other meta-learning frameworks (e.g. Proto-Net) in **Section 3.5** and with more details in **Appendix A.2**.
>
> **Requested Changes 1:** Many thanks to the reviewer's suggestion, as mentioned in our response to **Weakness 1**, we have included in the additional results for Omniglot and will include results of mini-Imagenet once they are available, along with the result discussion, in **Appendix G** of the revised paper.
>
> **Requested Changes 2:** As explained in our response to **Weakness 2**, we have represented or even extended MAML to the three supervised baselines in the paper: **Supervised-Original* (which is the original MAML), **Supervised-All**, and **Supervised-Oracle**, throughout the experiments of the paper.

---

> > ### Comment · Reviewer_DHBP · 2026-04-06
> >
> > Thank you for your response to my review. I am happy that both acceptance criteria for TMLR have been met, and I will recommend acceptance.

---

### Review · Reviewer_ELWT · 2026-03-17

**Summary Of Contributions:**

This paper revisits the effectiveness of meta-learning for few-shot learning and attributes its underperformance to limited task diversity in existing benchmarks. It proposes DRESS, a disentangled representation-based self-supervised meta-learning framework that generates diverse training tasks to improve adaptability. Experimental results demonstrate that DRESS consistently outperforms existing methods across datasets with varying factors of variation and complexity, highlighting the importance of task diversity in meta-learning.

**Audience:**

Yes

**Audience Explanation:**

Yes. The findings are relevant to researchers working on meta-learning, representation learning, and few-shot learning, and would be of interest to a meaningful portion of the TMLR audience.

**Claims And Evidence:**

Yes

**Claims Explanation:**

Yes. The claims are generally supported by empirical results across multiple datasets with varying complexity, and the experimental design is reasonably clear and consistent with the stated objectives.

**Requested Changes:**

1. The central claim that meta-learning underperforms due to insufficient task diversity is interesting but not fully substantiated. The paper would benefit from a more direct and quantitative analysis (e.g., controlled experiments varying task diversity, or metrics characterizing task heterogeneity) to validate this hypothesis.

2. The description of how disentangled representations are used to generate self-supervised tasks is somewhat abstract. More detailed explanations, possibly with concrete examples or visualizations of the disentangled factors and constructed tasks, would improve clarity.

3. The paper lacks detailed ablation experiments to isolate the contribution of each component (e.g., disentanglement module, task construction strategy, meta-learning objective).

4. While the paper positions itself against simple pre-training approaches, the experimental comparison with recent strong pre-training or fine-tuning baselines is limited.

5. The paper does not discuss the computational overhead introduced by disentangled representation learning and task generation. Providing runtime, training cost, or scalability analysis would be useful for practitioners.

---

> ### Author Response · Authors · 2026-03-24
> **Additional experiments introduced & Clarification on detailed experiments and analysis**
>
> We sincerely thank the reviewer's evaluation and comments. Please find below our responses to each of the requested changes.
>
> **Point 1:** We totally agree on the necessity of quantitative analysis into the task diversity and its effect on the few-shot adaptation performances. Therefore, under **Section 3.6**, we have proposed a quantitative metric for task diversity. Furthermore, we have provided the correlation plot between our quantified task diversity and the learning performance under **Section 5.4**. Many thanks to the reviewer's comment regarding controlled experiments varying the task diversity, we have further included experiments on the conventional low-task-diversity benchmarks, namely omniglot (and miniImagenet, for which we are still running experiments, and will provide results ASAP), in our revised paper.
>
> **Point 2:** The detailed procedure on task construction based on disentangled representations is listed step-by-step under **Section 3.1-3.4**, as well as illustrated in **Figure 1**. Furthermore, for concrete examples and visualizations regarding such tasks created by DRESS, we refer to **Figure 3, 4** under **Section 5**, and **Figure8, 9** under **Appendix G**. Within the caption for each of these figures, we described on the exact factor of variation that the task focuses on.
>
> **Point 3:** We fully understand the importance of ablation experiments, and therefore we have dedicated **Section 5.3** for ablation studies covering each major design element within the DRESS pipeline. Specifically, we have conducted ablation experiments focusing on the importance of disentangled representations VS regular latent representations, the explicit alignment stage after obtaining the disentangled representations, as well as the clustering strategy on the representations for task construction. The reviewer brought up a good point on the meta-learning objective. For this paper, throughout all the meta-learning methods including the baselines, we have adopted the conventional meta-learning objective as the task performance averaged across the meta-training set, which in this case, is the few-shot learning loss (CE loss). While there could potentially be possible alternatives for the meta-learning objective used to train, this objective is the most straight-forward and predominant choice throughout the literature on meta-learning or few-shot learning.
>
> **Point 4:** Actually, besides the most obvious baseline MAML and PTFT, we have also included two very recent meta-learning baselines: **Meta-GMVAE** and **PsCo**. These two baselines represent two main trends in the meta-learning researches: meta-learning under the framework of probabilistic inference (Meta-GMVAE) and meta-learning coupled with contrastive learning (PsCo); and present the state-of-art results not only in their respective direction, but among meta-learning approaches in general. With these two methods included, the list of baselines we compare DRESS to composes a strong line-up covering a wide spectrum of methods for few-shot learning.
>
> **Point 5:** Actually, in **Appendix E**, we have provided both the hardware setup, as well as the computation time of DRESS, together with each meta-learning baselines. Essentially, DRESS achieves its top performance at minimal time or computational overhead. This is thanks to the fact that DRESS does not introduce additional heavy computational step, with the encoding stage being present across all the baselines, and our latent dimension alignment stage involving simple non-parametric calculations (i.e. K-Means). Therefore, DRESS enjoys great scalabilities and would not fall behind of other competing methods on large datasets as we have shown in our experiments.

---

### Decision · Action_Editor_GV1B · 2026-04-22

**Recommendation:** Reject

**Audience:**

Yes

**Audience Explanation:**

The topic and findings would be of interest to researchers in meta-learning and few-shot learning.

**Claims And Evidence:**

No

**Claims Explanation:**

While empirical results on several high-diversity benchmarks demonstrate the effectiveness of the proposed method, this paper lacks sufficient clarity and convincing evidence to support its central claims, as described in the followings:
- The methodology is described at a relatively abstract level, and key components such as disentanglement, clustering, task construction, and their interaction with the meta-learning objective are not clearly specified, making it difficult to fully understand and reproduce the approach.
- The empirical analysis is not sufficiently thorough to validate the main hypothesis, particularly regarding the role of task diversity, and does not clearly explain when or why existing methods fail. In addition, important evaluations are missing or incomplete, including standard benchmarks and key baselines, which limits the strength of the empirical evidence.

Overall, a major revision is required to address these issues.

**Resubmission Of Major Revision:**

The authors may consider submitting a major revision at a later time.